# Dynamical reconstruction of the upper-ocean state in the Central Arctic during the winter period of the MOSAiC Expedition

Ivan Kuznetsov[a], Benjamin Rabe[a], Alexey Androsov[a], Ying-Chih Fang[b], Mario Hoppmann[a],
Alejandra Quintanilla-Zurita[a], Sven Harig[a], Sandra Tippenhauer[a], Kirstin Schulz[c], Volker Mohrholz[d],
Ilker Fer[e], Vera Fofonova[a], and Markus Janout[a]

[a]Alfred-Wegener-Institut Helmholtz-Zentrum für Polar- und Meeresforschung, Bremerhaven, Germany
[b]National Sun Yat-sen University University, 80424 Kaohsiung, Taiwan
[c]Oden Institute for Computational Engineering and Sciences, The University of Texas at Austin, Austin, TX, United States
[d]Leibniz-Institute for Baltic Sea Research Warnemünde, Germany
[e]Geophysical Institute, University of Bergen and Bjerknes Center for Climate Research, Bergen, Norway

**Correspondence:** Ivan Kuznetsov (ivan.kuznetsov@awi.de)

**Abstract.** This paper presents a methodological tool for dynamic reconstruction of the state of the ocean, based, as an example, on observations from the Multidisciplinary Observatory for the Study of Arctic Climate (MOSAiC) experiment. The data used in this study were collected in the Amundsen Basin between October 2019 and January 2020. Analyzing observational data to assess tracer field and upper-ocean dynamics is highly challenging when measurement platforms drift with the ice pack due
to continuous drift speed and direction changes. We have equipped the new version of the coastal branch of the global finite volume sea-ice ocean model FESOM-C with a nudging method. Model nudging was carried out assuming a quasi-steady state. Overall, the model can reproduce the lateral and vertical structure of the temperature, salinity, and density fields, which allows for projecting dynamically consistent features of these fields onto a regular grid. We identify two separate depth ranges of enhanced eddy kinetic energy located around two maxima in buoyancy frequency: the depth of the upper halocline and the
depth of the warm (modified) Atlantic Water. Simulations reveal a notable decrease in surface layer salinity and density in the Amundsen Basin towards the north but no significant gradient from east to west. However, we find a mixed layer deepening from east to west, with a 0.084 m/km gradient at 0.6 m/km standard deviation, compared to a weak deepening from south to north. The model resolves several stationary eddies in the warm Atlantic Water and provides insights into the associated dynamics. The model output can be used to further analyse the thermohaline structure and related dynamics associated with
mesoscale and submesoscale processes in the Central Arctic, such as estimates of heat fluxes or mass transport. The developed nudging method can be utilized to incorporate observational data from a diverse set of instruments and for further analysis of data from the MOSAiC expedition.

## 1 Introduction

Mesoscale and submesoscale eddies have been observed in many of the world's oceans, including the Arctic. The fluxes driven
by those eddies play a significant role in Arctic Ocean dynamics, such as the ventilation of the halocline and the transport of organic and inorganic matter (Dmitrenko et al., 2008; Meneghello et al., 2021; Marcinko et al., 2015; Pnyushkov et al., 2018;

Mahadevan, 2016; Mahadevan et al., 2010; Gula et al., 2022; Nishino et al., 2018; Watanabe, 2011). Most studies on under-ice eddy dynamics in the Arctic focus on the ice edge or the shelf break zone, where the eddy activity is maximal. Still, little is known about eddies in the ice-covered central Arctic Ocean.

Any mesoscale activity is inevitably linked to submesoscale dynamics, such as filaments around mesoscale eddies (e.g. Della Penna and Gaube, 2019; Mahadevan, 2016). The intensity of mesoscale dynamics can be represented by the eddy kinetic energy (EKE), found to be stronger in areas with low sea ice concentration (von Appen et al., 2022) and weakened by friction under sea ice, e.g. in the Arctic interior and/or in winter (Meneghello et al., 2021). Vertical eddy diffusivity of mass and heat, and associated fluxes, as well as vertical velocity, can be enhanced significantly by under-ice eddies (Manucharyan and
Thompson, 2017); conversely, submesoscale flows can both enhance those fluxes and suppress them by restratifying the mixed layer (Mensa and Timmermans, 2017). Observations and simulations have shown that the occurrence of eddies varies on monthly (Pnyushkov et al., 2018) to interannual (Zhao et al., 2016) time scales, and individual eddies may persist up to several months (Scott et al., 2019). Despite the intensification of eddy dynamics at the basin boundaries, von Appen et al. (2022) note that in the central Eurasian Basin, EKE is significant in both the halocline and the Atlantic Water layer, according to
high-resolution simulations.

Eddy dynamics not only impact ocean physics but also biochemical and ecosystem processes in the Arctic Ocean. Llinás et al. (2009) illustrate the possible mechanism of zooplankton transport from the Chukchi Shelf into the interior Canada Basin by eddies. O'Brien et al. (2013) find a significant role of eddies on the particle flux, and Oziel et al. (2022) demonstrate in a modelling study that lateral submesoscale eddy transport is one of the dominating processes controlling the nitrate supply in
the central Arctic Ocean. Omand et al. (2015) show that small-scale vorticity could be responsible for high particulate organic carbon concentrations in deeper layers. According to their calculations, submesoscale, eddy-driven fluxes can contribute as much as half of the total springtime export of particulate organic carbon from the upper ocean to deeper layers in the subpolar region.

Despite ongoing efforts to develop and improve climate models, accurately representing mesoscale and submesoscale dy-
namics remains challenging. Androsov et al. (2020) compared ocean models with various horizontal resolutions to observed ocean bottom pressure and found only a modest correlation between the models and observations. They conclude that high resolution is necessary for areas with high mesoscale eddy activity. According to Zhao et al. (2014), the radii of observed halocline mesoscale eddies are on the order of a few kilometres in the Arctic Ocean, where the first baroclinic Rossby deformation radius varies between 1 to 15 km (Nurser and Bacon, 2014). The necessity to resolve such small scales makes it challenging to
explicitly model mesoscale and submesoscale features in global climate models due to the lack of horizontal resolution. Recent developments in the ocean and coupled climate models, software, and hardware give a possibility for simulations with a very high horizontal resolution, for example, Wang et al. (2020); Maslowski et al. (2008); Regan et al. (2020); Lyu et al. (2022); Hordoir et al. (2022).

Most studies on eddy dynamics in the Arctic have focused on the marginal ice zone or coastal currents, the main limitation
of mesoscale and submesoscale research in the central Arctic being observational data. Unfortunately, standard methods for observing eddies, such as satellite remote sensing or glider and transect campaigns, have been challenging in near-perennially

ice-covered seas. To fill this gap, the Multidisciplinary Observatory for the Study of Arctic Climate (MOSAiC, e.g. Rabe et al., 2022) designed a Distributed Network (DN; Rabe et al., 2024) of autonomous ice-tethered systems (hereafter referred to as "buoys") around the MOSAiC Central Observatory (CO). The CO included a series of buoys and mostly manual ocean

observations at the R/V *Polarstern* and a site about 300 m away from the ship ("Ocean City"; OC). This setup enabled us to obtain regular, non-autonomous measurements during the MOSAiC expedition. Many extremely diverse observations, from point measurements to profiles, differ significantly in temporal frequency, from irregular weekly measurements to measurements every 2 minutes, and in spatial resolution, from tens of kilometres to tens of meters. Moreover, looping the DN drift trajectory and the intersection of the trajectories of different buoys add complexity to the observed data. Nevertheless, using

relatively simple data analysis methods, several mesoscale eddies have been identified in the Amundsen Basin (Hoppmann et al., 2022; Fang et al., 2023; Quintanilla-Zurita and et al., (in prep.). However, these methods have limitations in showing the overall three-dimensional picture due to the above-mentioned complex drift trajectories, and analyzing the buoy measurements spatially can be challenging, particularly in estimating lateral gradients of tracers or velocity shear. This poses the question of best analysing such scattered data and dynamics and their role in vertical transport. One possible approach is to use interpo-

lation techniques, such as optimal interpolation (Bretherton et al., 1976) or Data Interpolation Variational Analysis (Troupin et al., 2012; Barth et al., 2014). However, the lateral scales of phenomena on interpolated maps can be limited by distances between the observing buoy systems or parameters of the interpolation algorithm rather than physical processes.

Reconstructing temperature, salinity, and density fields with a model by data assimilation allows estimating dynamically consistent lateral features of these fields on a regular grid. Androsov et al. (2005) and Rubino et al. (2007) used in situ obser-

vations with three-dimensional non-hydrostatic modelling to investigate the non-stationarity of the dynamics and evolution of mesoscale chimneys in the Greenland Sea. Together with an analytical solution, this allowed the authors to investigate these eddies' inertial pulsations, shape, and velocity structure, as well as their significant effect on open-ocean deep-penetrating convection. Assimilating high-frequency variability data presents its significant challenges: First, the assimilation time (usually once every 10 days or even daily averaging in extreme cases) significantly increases computation time. Second, the assimila-

tion process involves averaging over a significant data radius, resulting in a smoothing effect on the assimilated data (Androsov et al., 2018). The nature of the data has to be considered when employing advanced methods such as the four-dimensional variational method (Courtier et al., 1994; Mogensen et al., 2009) or the Parallel Data Assimilation Framework (Nerger et al., 2020). Unfortunately, the high-frequency variability and the scales of the observations inherent in the MOSAiC data make it impractical to apply these methods to eddy analysis. Alternatively, nudging has several advantages for ocean data assimilation,

including its ease of implementation in complex numerical models, low computational demands, and the smoothness of the solution over time (Ruggiero et al., 2015).

Our study aims to extend current knowledge of eddy dynamics in the central Arctic by using the three-dimensional regional model FESOM-C with very high vertical (up to 1 m) and horizontal (up to 250 m) resolution. We utilize observed temperature and salinity data from the MOSAiC DN buoys as part of the forcing for the numerical model, employing a nudging method

with a quasi-steady-state approximation. Our objective is to present a newly developed modelling tool to reconstruct gridded

fields of water properties based on MOSAiC DN observational temperature and salinity. Additionally, we aim to estimate the properties of mesoscale and submesoscale dynamics and their potential variability during the MOSAiC expedition.

This paper is organized as follows. Section 2 presents the numerical model, observations, new nudging methodology, and experimental design. In section 3, we present the results of the simulations and model validation. The analysis of (sub)mesoscale dynamics and distribution of eddy kinetic energy from the reconstructed dynamical fields are discussed in Section 4. In section 5 we summarize the results.

## 2 Methods

### 2.1 Observational data

Here, we use an observational data set obtained as part of the physical oceanography work program during the field phase of the Multidisciplinary Observatory for the Study of Arctic Climate (MOSAiC) in 2019/20 (Shupe et al., 2020). A description of the physical oceanography part of the experiment with a general description of the instruments and methodology is presented in Rabe et al. (2022). Various instruments obtained temperature and salinity observations used in this work: most of the data was measured by autonomous ice-tethered systems ("buoys") within the DN originally deployed by the icebreaker *Akademik Fedorov* (Krumpen and Sokolov, 2020) radially around the icebreaker *Polarstern* tethered to the sea ice at the Central Observatory. Eight buoys termed "Salinity-Ice-Tether" (SIT) measured temperature, conductivity, and pressure, with derived salinity and depth, at five depths of 10, 20, 50, 75, and 100 m with a sampling interval of 2 to 10 minutes, with a distance between subsequent data points as small as 80 m. The sensors used on these buoys have an initial accuracy of $\pm 0.003$ mS cm$^{-1}$ for conductivity, $\pm 0.002$ °C for temperature, and $\pm 0.1\%$ of the full range for pressure. The sensor stability rating is 0.003 mS cm$^{-1}$ and 0.0002 per month for conductivity and temperature, respectively, with a yearly rating of 0.05% of the full scale for pressure. The detailed description of the instruments and the data obtained are given by Hoppmann et al. (2022). We further used data from three Ice-tethered profilers (ITP; Toole and Krishfield, 2016). The time between subsequent profiles varied from several hours to days, depending on the specific system. The maximum depth reached by these profilers was about 700 m, with the minimum depth varying from 5 to 8 m. Thus, the measurements with profilers cover depths in the warm waters of Atlantic origin (referred to as "warm Atlantic Water") and beyond. A total of 1114 profiles were used to nudge the model. The nudging process included additional profiles from CTD/rosette casts at the Central Observatory, both from *Polarstern* (PS-CTD; 25 profiles with depths range from 2 to up to 4450 m) (Tippenhauer et al. (2023a)) and from a location a few hundred meters away from the ship, at the "Ocean City" (OC-CTD; 44 profiles with depth range from 2 up to 500 m) (Tippenhauer et al. (2023b)). ITP profiles with unstable stratification or a vertical range of less than 10 m were excluded from the analysis. Data from the profiles were averaged with a standard pressure interval of 1 dbar (a depth interval of about 1 m) as indicated in the data sources. All devices' measurement accuracy is much higher than the error introduced by interpolation and the nudging scheme.

The observations used for model nudging covered the region between 87.6 ° East, 139.5 ° East, $84.5°$ North and $87.5°$ North, corresponding to the MOSAiC drift from October 2019 to January 2020. During this period, the MOSAiC expedition drifted

from the southeast to the northwest. During the initial phase of the drift, the trajectories of the DN buoys, shown in Figure 1 by
colored dots, exhibited predominantly straight paths. The later part is characterized by the presence of overlapping loops in the
trajectories when the regional sea ice cover changed drift direction. These loops of the trajectories of different measurement
platforms increase the area of the data coverage compared to the straight drift while, at the same time, introducing uncertainty
in the spatio-temporal interpretation of the data. Measured parameters could differ between data measured at the same position
at different times, leading to the aliasing of the observed signal. The average ice drift speed during the observation period was
12 cm/s.

To validate the model results, we used independent temperature and salinity data from a turbulence microstructure profiler
(MSS; Schulz et al., 2022). The model did not use these data for nudging. MSS profiles were obtained at Ocean City, on a
near-daily resolution, in sets of at least 3 profiles. The profiles are averaged to 1 m vertical resolution and corrected against CTD
profiles, calibrated with water samples. For comparison to the model fields, we used 305 profiles (see black crosses in Figure 1).
The MSS data, while not included in the nudging process and thus considered independent to a degree, inevitably exhibits some
degree of autocorrelation with DN and PS measurements. This is particularly due to the spatial distances between DN buoys
and the temporal and spatial dispersion of data from PS and MSS. Consequently, we acknowledge the data as independent with
the caveat that a certain level of autocorrelation is present, reflecting the inherent spatial and temporal structures within the
observational network.

## 2.2 FESOM-C model

The FESOM-C model used in this work (Danilov and Androsov, 2015; Androsov et al., 2019) is a coastal branch of the global
Finite volumE sea ice–Ocean Model (FESOM2) (Danilov et al., 2017). In addition to the partially common interface of the
models, FESOM-C has specific features that are important for our work. The model was originally developed for applications
with a high horizontal resolution as fine as several meters (Neder et al., 2022; Kuznetsov et al., 2020; Fofonova et al., 2019).
This model uses the discretization of cells and vertices of a finite volume, which allows the use of unstructured computational
grids. We use this function to move the boundary of the computational domain away from the region of interest, the "core"
of the model grid, without creating a system of nested grids. At the same time, the horizontal resolution outside the core is
quite coarse, which allows us to reduce the influence of the boundary on our solution inside the core. The most important
distinguishing feature from the global FESOM2 model is the possibility of using hybrid grids consisting of triangles and
squares. This approach's effectiveness in enhancing stability and using larger time steps is shown by Danilov and Androsov
(2015) and Androsov et al. (2019). Additionally, this model branch uses sigma layers in the vertical direction.

The parallel Algebraic Recursive Multilevel Solver (pARMS, (Li et al., 2003b)) used in FESOM2 was used to calculate
the sea level using a semi-implicit method. Since we study the processes in the deep-water region, where the effect of bottom
friction is minimal, and the barotropic mode does not play a key role, we modified the scheme to a semi-implicit calculation of
the sea level, omitting the solution of the block of average equations.

The thermodynamics of the sea ice model component has not been used in the current work. Alteration of the ocean surface
temperature and salinity due to ice formation and melting has been implemented through model nudging towards observational

data. The minimum depth of the observational data from instruments ranged from 2 to 10 m. Considering that the mixed layer depth exceeded 20 m, the temperature and salinity within the mixed layer were well represented in the data. The effect of sea ice presence on the dynamics of the ocean surface layer has been parameterized by the friction between ice and ocean. Thus, we do not consider the additional momentum transfer due to ice drift. The effect of ice drift has been accounted for in the turbulence closure and is described in the following section.

In contrast to previous publications, we have implemented parallel calculations based on the Message Passing Interface (MPI). For the dynamic part of the model, the MPI scheme is similar to that of FESOM2 but applied to hybrid grids. In contrast to the global model, the organization of parallel output and input for boundary conditions at open surface boundaries were written using the PnetCDF library (Li et al., 2003a). This made it possible to take advantage of the flexibility of the previous openMP FESOM-C I/O version.

## 2.3 Turbulence closure

The turbulence closure equation based on the Prandtl-Kolmogorov hypothesis described in Androsov et al. (2019) calculates turbulent vertical flows. Compared to the original version of the FESOM-C model, the modification of this equation concerned only the parametrization of the turbulence scale $l$. The need for this change is associated primarily with the parametrization of the ice-water layer and a more dynamic description of the moving mixed layer (ML). At the preliminary stage, an average upper bound ice-drift velocity is estimated at 0.7 m/s, which is used as an upper boundary condition for the dynamic wind speed in the turbulent energy budget equation. This parameter is used across the entire domain and throughout the entire period of the model's nudging. Since we use a quasi-steady-state approximation (see nudging section 2.5), this parameter remains unchanged throughout the computation process but does not represent individual storm or lead events. We compensate for these with model nudging to observations. In the second stage, the thickness of the ML ($h_{ml}$) is estimated as the depth at which the practical salinity increases by 0.5 from its surface value. Estimates of $h_{ml}$ less than 20 m are set to 20 m. This is one of the commonly used definitions of the ML depth. The exact definition of ML depth does not play a crucial role in our task. Then, the scale of turbulence in the upper ML is determined by

$$l = \frac{\kappa}{h_{ml}} \cdot Z_H \cdot Z_\zeta, \tag{1}$$

where $Z_H = z + h_{ml}$, $Z_\zeta = z + z_\zeta$, $\kappa \sim 0.4$ is the von Kármán constant, $z$ is the depth, positive downwards, and $z_\zeta$ is the roughness parameter for the ice-water layer. Underneath the surface ML, $h_{ml}$, the scale of turbulence is given by:

$$l = \frac{\kappa}{H - h_{ml}} \cdot Z_H \cdot Z_{ml} \cdot C, \tag{2}$$

where $H = h + \zeta$ is the full water depth, $Z_{ml} = -z + (H - h_{ml}) + z_b$, $z_b$ is the roughness parameter for the bottom, and the constant $C \sim 0.01$ is set to reduce the scale of turbulence underneath the ML. This approach to determining the scale of turbulence ensures the smoothness and minimization of turbulent exchange at the boundary of the ML and the water column underneath the ML.

## 2.4 Model domain

The model domain is a parallelepiped in Cartesian coordinates with a solid boundary. Since the FESOM-C model allows computing on mixed unstructured meshes, we can adjust the spatial resolution without using nested grids. For the final computations, two spatial configurations of the model are used. The general conditions for these configurations were that, near the horizontal boundaries of the domain, the spatial resolution was relatively coarse, about 5 km. Such a coarse resolution at the boundaries serves two purposes: First, absorption due to the stronger dissipation of the uncertainty of the boundary informa-

tion and its levelling to calculations in the model's core is the most important in our application. Second, there are significant savings in computational resources. The first configuration of the model has a resolution of up to 1 km in the area of interest and contains 96000 nodes. This setup is mainly used at the initial modeling stage to form the initial conditions for a spatially detailed configuration of the study domain. For the coarse-resolution initial condition, a single profile was applied throughout the entire model domain; details are provided in 2.6. The second configuration has a resolution of up to 250 m at the model's

core and contains about 1.3 million nodes (see Figure 1). This setup is used for the final computations and data analysis. A more detailed description of the experiments on these two grids is presented in section 2.6 and Figure 2. The vertical structure is the same for both configurations and contains 240 vertical $\sigma$-layers. The model domain covers the entire water column, reaching a maximum depth of 4450 m, representing this region's average depth. At the same time, the upper layer up to 150 m has an effective resolution of up to one meter, which makes it possible to significantly improve the representation of the vertical

aspects of the submesoscale dynamics of surface ML compared to global models. The model domain is about 660 by 525 km and limited geographically to between 90° East and 140° East longitude as well as 83° North and 87.7° North latitude.

## 2.5 Nudging

Nudging, along with the specifics of its application, plays a key role in our study. Model nudging was conducted under the assumption of a quasi-steady state, ensuring that the model was nudged with all observational data simultaneously. The model

does not consider the time the observations were taken, which is a reasonable approximation at high drift speed relative to the water velocity. At the same time, we assume no significant relation of the submesoscale baroclinic structure at different ends of the model domain. Thus, we obtain a quasi-stationary solution by nudging the model to observations that are not separated in time. This approach can be viewed as the outcome of dynamically justified interpolation. In analyzing the obtained fields, it should be remembered that the nudging data spans four months. This duration impacts various system parameters, such as

the depth of the mixed layer. Atmospheric influences on the flows are captured solely through nudging, with the data density shaping a smoothed pattern. Given the months-long temporal span of the observational data in a quasi-stationary setting, this method may introduce horizontal gradients in the temperature and salinity fields. The caveats of this approach are further discussed in section 4.3.

We applied a simple nudging algorithm: This method adds a nudging term to the evolution equation proportional to the

difference between the model temperature and salinity and the observational data at a given location. The full formulation of

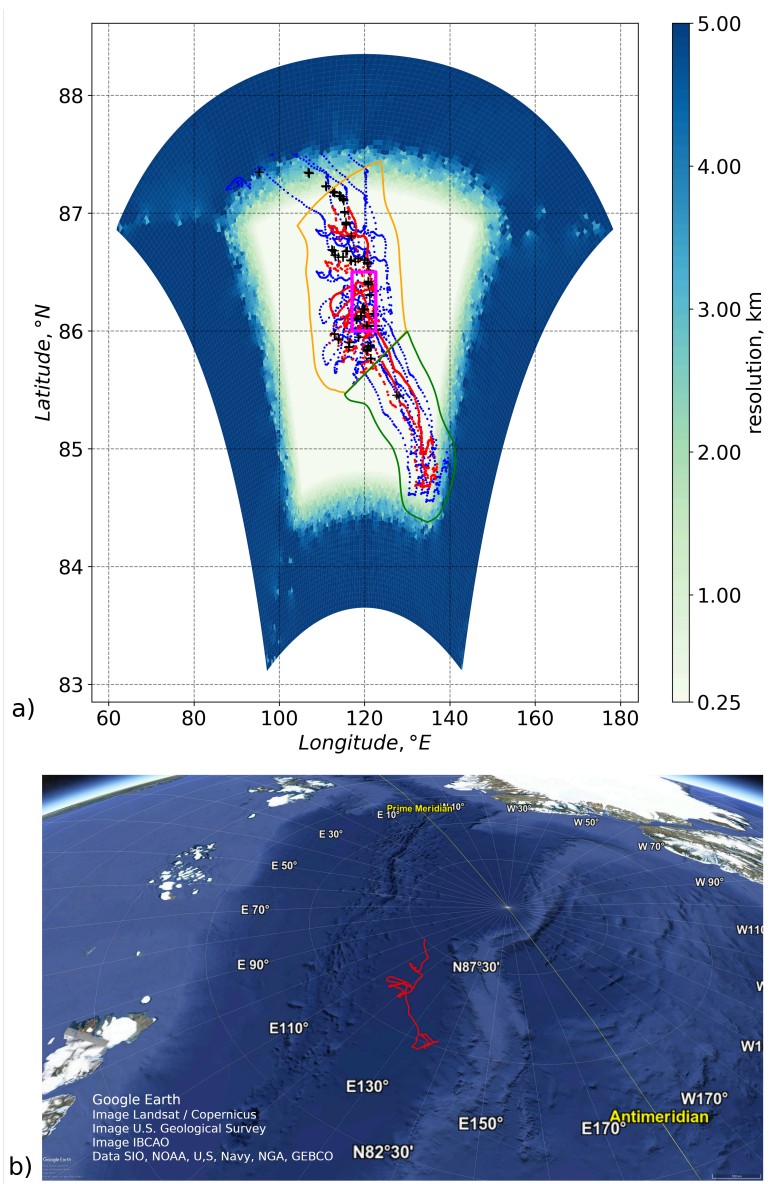

**Figure 1.** a) - Model domain resolution with the position of the observational data used in this study, covering a period of ≈2.5 months. The positions of the observational data used for nudging are separated by instrument type: blue - 5 SIT buoys, with CTDs at 10,20,50,75,100 m depth (Hoppmann et al., 2022). red – PS- and OC-CTD casts (Tippenhauer et al., 2023a, b), and ITP profiles (Toole and Krishfield, 2016). Black crosses - position of MSS profiles used for validation Schulz et al. (2022). The green polygon indicates an area with mainly straight drift trajectories, whereas the orange polygon indicates an area with often overlapping buoy trajectories. The magenta rectangle shows the area of Figure 11. b) - The border situation map with the ship's drift trajectory is marked in red. ©Google Earth 2019

the model is given in Androsov et al. (2019). Here, we present only the tracer equations in which changes have been made and where the last term represents nudging,

$$\frac{\partial \Theta_j}{\partial t} + \frac{\partial}{\partial x_i}(\mathbf{u}_i \Theta_j) + \frac{\partial}{\partial z}(w \Theta_j) = \frac{\partial}{\partial z} \vartheta_\Theta \frac{\partial \Theta_j}{\partial z} + \nabla_2 (K_\Theta \nabla_2) \Theta_j + C_k (\Theta_{oj} - \Theta_j) \tag{3}$$

Here, $i = 1, 2$, where $x_1 = x$ and $x_2 = y$ correspond to the spatial coordinates, and $u_1 = u$ and $u_2 = v$ represent the components
of a vector field, in these coordinates. Summation over the repeating indices $i$ and $j$ is implied. Additionally, $j = 1, 2$ is used where $\Theta_1 = T$ represents the potential temperature, and $\Theta_2 = S$ represents salinity, respectively. $\nabla_2$ is a two-dimensional gradient operator. $\vartheta_\Theta$ and $K_\Theta$ are the corresponding vertical and horizontal diffusion coefficients. $\Theta_{oj}$ is the observational data interpolated on mesh (see below). $C_k$ is the spatiotemporal relaxation coefficient different for different sources of observed data, and $k = 1, 2$ represents the point sources (SIT) and profiles (ITP, PS-CTD and OC-CTD). The term responsible for
nudging was included only for grid nodes near observations. To do this, a mask of nodes has been precalculated for each type of observation, which is explained below. While this nudging method breaches the principle of continuity, its use is limited to distinct observational sites rather than uniformly applied across the entire area. This focused approach helps prevent significant issues when setting initial conditions for a free simulation.

The observed data were separated into two groups divided by the nature of these data: The first group of data was obtained
using the SIT buoys. Data from these buoys have a high temporal resolution of up to 2 minutes and a horizontal spatial resolution of up to 80 m, both high compared to the temporal and spatial resolution of ITP profiles. At the same time, each buoy provided data from a maximum of 5 different depths. The second data group comprises profiles obtained from ITP and PS/OS STD instruments.

$\Theta_{o1}$ was precalculated, and the data from the SIT buoys were interpolated onto the computational grid. Interpolation was
made for two-dimensional fields for the corresponding depths of SIT buoy sensors 10,20,50,75 and 100 m. We used a modified inverse distance weighting method (Shepard, 1968) combined with fast spatial search structure kd-tree (Maneewongvatana and Mount, 1999). Interpolation was done within a maximum distance of 1 km from each observation position and a maximum number of 30 grid nodes affected by particular measurements (Figure 2 a). The rest of the mesh nodes were masked as nodes that did not participate in nudging. These up to 4 model nodes are affected by one particular observation for the model mesh
with 1 km resolution. At the same time, one particular measurement affects the surrounding model mesh nodes up to 750 m away for the mesh with 250 m resolution due to a limit of a maximum of 30 nodes. When multiple data values were present, such as when the buoy trajectories intersected, a weighted average was calculated for a particular grid point (see Figure 2 b)). The model then used the interpolated 2-D fields for nudging. The model nudges the simulated fields to the observation fields for each of the five depths with a spread of 3 m from the observation depths (see Figure 2 c)). Thus, the spatiotemporal relaxation
coefficient for SIT buoys takes the form:

$$C_1 = \begin{cases} T_{relax}/(1 + 0.25 \cdot |Z_{obs} - Z|^2) & : |Z - Z_{obs}| \leqslant 3 \\ 0 & : |Z - Z_{obs}| > 3 \end{cases} \tag{4}$$

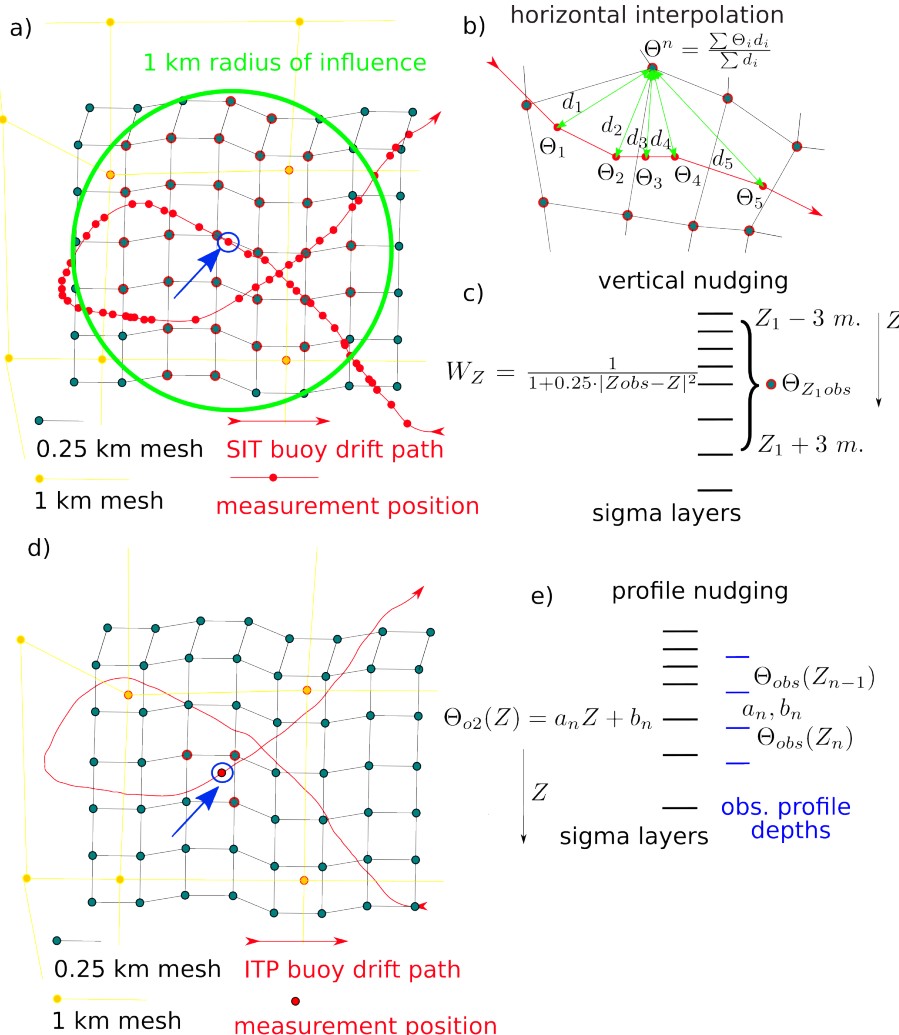

**Figure 2.** a), b) and d) - Schematic of interpolation and masking on the mesh. Blue and yellow circles are vertices of 0.25 and 1 km resolution meshes. The red lines and circles represent the paths and measurement positions of SIT buoys in a) and of ITP and PS/OC CTDs in d). c) - vertical relaxation weights distribution. e) - vertical interpolation of observed profile. $a_n$ and $b_n$ linear interpolation coefficients. Meshes vertices marked in red are influenced by the measurement marked with a blue circle.

where $T_{relax}$ is the temporal relaxation coefficient equal to $1.1574 \cdot 10^{-5}$ 1/s (one day), $Z$ is a depth of the sigma layer, $Z_{obs}$ is one of five depths of the CTDs at the SIT buoys. Thus, the model's nudging occurs near the observation point +-3 m, but the strength of the nudging decreases with distance along the vertical from the observation point.

As a sensitivity study, a larger number of possible maximum values of neighbouring nodes was also used. However, this does not significantly affect the final result. The total number of measurements was about 630000 for each parameter of salinity and temperature.

In contrast to the SIT buoys, the model was nudged to the profiles by the ITP profilers and the PS- and OC-CTD only at the three nodes closest to the observation position (see Figure 2 d)). If more than one profile belonged to one node, then, as in the case of SIT buoys, the inverse distance weighting method was used to average the profiles. The remaining nodes of the computational grid did not participate in nudging. Vertically, the model was nudged only at the horizons where the data from the profiles were present. Linear interpolation coefficients $a(z)$ and $b(z)$ were precalculated for each cell between the standard depths where observed profiles have data (see Figure 2 e)). The model only reads the interpolation coefficients and reconstructs the measured values at each model depth. This approach adds flexibility in setting the vertical arrangement of model sigma layers and avoids data interpolation in model calculations. In this way, the spatiotemporal relaxation coefficient for profiles was $C_2 = T_{relax}$ and $\Theta_{o2} = a(z)z + b(z)$.

The significant difference in the radius of influence of the data arises from the horizontal resolution of the measurement data from the first and second groups (see prior definition). An increased radius for the SIT buoys is necessary to smooth the fields when crossing the trajectories of the buoys. In the case of a small radius, this leads to artificial fronts. In the case of profiles, the probability of finding measurements at different times in one place is extremely small due to the low-frequency sampling of these instruments. SIT buoys provide data every 2 minutes, offering high-frequency observations. In contrast, instruments that record incomplete vertical profiles do so once daily, while those capturing full-depth profiles from the ship CTD do so at most once a week. Consequently, the influence of these profiles on nudging the surface layers is notably less compared to the more frequent data provided by SIT buoys. At the same time, zones deeper than 100 m are determined exclusively by profiles. The dynamics activity and variability in the upper 100 meters of the ocean are significantly higher compared to deeper regions. The abundance of data in this upper layer allows for a detailed representation of submesoscale processes, leveraging the system's dynamic nature. Conversely, the deeper zones exhibit less variability, making them amenable to accurate representation with fewer data points. This differential data density aligns with the varying dynamical characteristics of these oceanic layers, ensuring the model's efficacy across depths. It's important to note that nudging can lead to a violation of the continuity principle. However, data nudging is restricted to specific observation locations rather than applied across the entire area. This localized application prevents significant issues from arising.

Several storm events were observed during the measurements, including one strong storm (Fang et al., 2023). Strong storms alter the dynamic nature of the surface layer and lead to the ventilation of the upper mixed layer (ML). In our quasi-stationary approach, the simulation results do not directly capture the dynamics during a storm. However, the model indirectly considers the effects of storms through nudging changes in temperature and salinity, albeit in a more smoothed manner.

## 2.6  Experiment description

The deep PS-CTD profile (PS122/1_10-44) conducted from *Polarstern* during November, from the surface down to the seafloor, was used as an initial condition for the whole model domain (see Figure 4 bold lines). To avoid instability in the initial conditions, the measured temperature and salinity of the surface layer were changed to constant values corresponding to a depth of 30 m.To reduce computing time, the initial run with nudging was conducted on the coarse mesh (1 km) described in Section 2.4 (see Figure 3). This "spin-up" simulation spans approximately 1 year of model time and ends at the point where

convergence for coarse resolution is achieved. In such a way, temperature and salinity differences between two successive time steps do not vary significantly. The resulting three-dimensional temperature and salinity fields from the "spin-up" simulation were then used as the initial conditions for the simulation with higher resolution (250 m). This high-resolution simulation, including nudging, lasted for 4 months of model time until convergence of the numerical solution was reached again.

Our nudging method violates the continuity principle and results in a disturbance of the velocity fields. To resolve this issue and satisfy continuity, an additional experiment was performed: A simulation with the high-resolution mesh and without nudging was run using as initial conditions the dynamical and tracer fields derived from the simulation with nudging, hereafter termed the "free run". The duration of the free run was 19 real days. Results of the free run are used to analyze the dynamical field, which is based on the nudged run and shows a structure similar to that constrained by the observational data. This approach reduces disturbances and violations of continuity, as described above.

In the following, we used the results from high-resolution mesh experiments, including the end of the nudging experiment and the free run after 2.5 days of free simulation, to compare these results with independent data. We also used the free-run at the 2.5-day time step to present the temperature and salinity reconstruction. The 2.5-day time step was chosen as sufficient time to distance from the moment when the last continuity violation occurred while ensuring the model did not drift too far from the observations. We exclusively used the free run at various time steps to analyse eddy dynamics and assess eddy kinetic energy.

## 3   Results

### 3.1   Model validation

The MSS data set described earlier was not used for the model nudging and serves here to validate the model with independent observations. Overall, the model can reproduce the lateral and vertical structure of the salinity and temperature fields, as represented by the independent MSS observations (Figures 5, 6 and 7a). Snapshots of the simulation with nudging and the free run after 2.5 days (Figure 5 b, c) show similar lateral salinity gradients in the ML. After 19 days in the free run (Figure 5 d) ), the lateral salinity gradient is smaller due to vertical and horizontal mixing. Salinity and temperature variability is slightly lower in the model compared to the observations (Figure 7c). The maximum deviation of the free run after 2.5 days from the observations is at the depths of the maximum vertical gradients in salinity (at about 37 m) and temperature (at about 150 m depth). Variability of salinity decreases with time in the free run when nudging is no longer taking on the role of external forcing. In the absence of nudging, the model tends to dissipate eddies and slump fronts, smoothing lateral gradients.

The statistics of the comparison between the model and SIT buoy data (Table 1) show that the model deviates from the observations despite nudging with observational data. The largest deviation is at the positions where buoy trajectories intercept each other. In such cases, the model points are aligned with at least two separate observations of the same variable at the same location, highlighting the limitations of the quasi-stationary approximation assumption. Typically, the model strives to replicate the smoothed values derived from these overlapping observations. Moreover, the horizontal resolution of the SIT buoy observations is often higher than the spacing of the model grid, which results in larger differences between individual

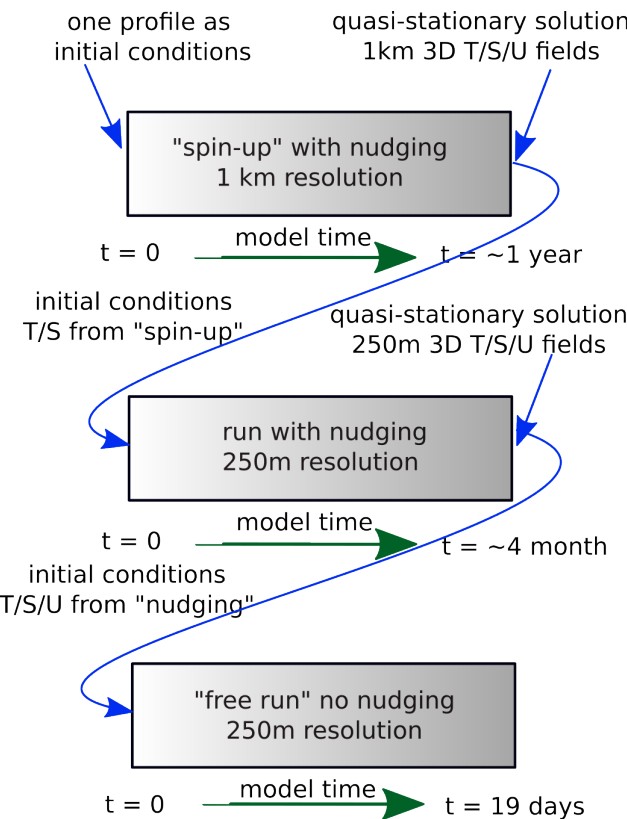

**Figure 3.** Schematic of conducted simulations. T/S/U are abbreviations for temperature, salinity, and velocity. The rectangles represent individual simulations.

observed values and model output, as one model grid point covers several observations in space. The root means square errors (RMSE) are in the range of the standard deviation (SD) of the model and the observations in the surface ML (Table 1). RMSE significantly decreases for salinity underneath the halocline and about half of the observed SD. At the same time, RMSE and SD increase for temperature underneath the halocline where temperature gradients increase (not shown).

In conclusion, following the model validation, our comparison with independent data indicates that our method yields sufficiently accurate results. Therefore, it can be reliably used for the reconstruction of three-dimensional fields.

### 3.2 T/S reconstruction

The modeled fields of free-run after 2.5 days, illustrated in Figure 6 through cross sections along 115°N and 86.2°E, show a decrease in ML salinity and density towards the north. The ML depth varies from about 36 in the south to 40 in the north with a minimum of 32 m, resulting in a gradient of 0.014 m/km with a 0.6 m/km standard deviation. Likewise, the ML's density changed by about 1.1 kg/m$^3$. Underneath the ML, isopycnal lines slope by more than 10 m along the section with less smaller-scale variability than at the bottom of the ML. In the west-east direction, the ML shoals amid less small-scale variability than

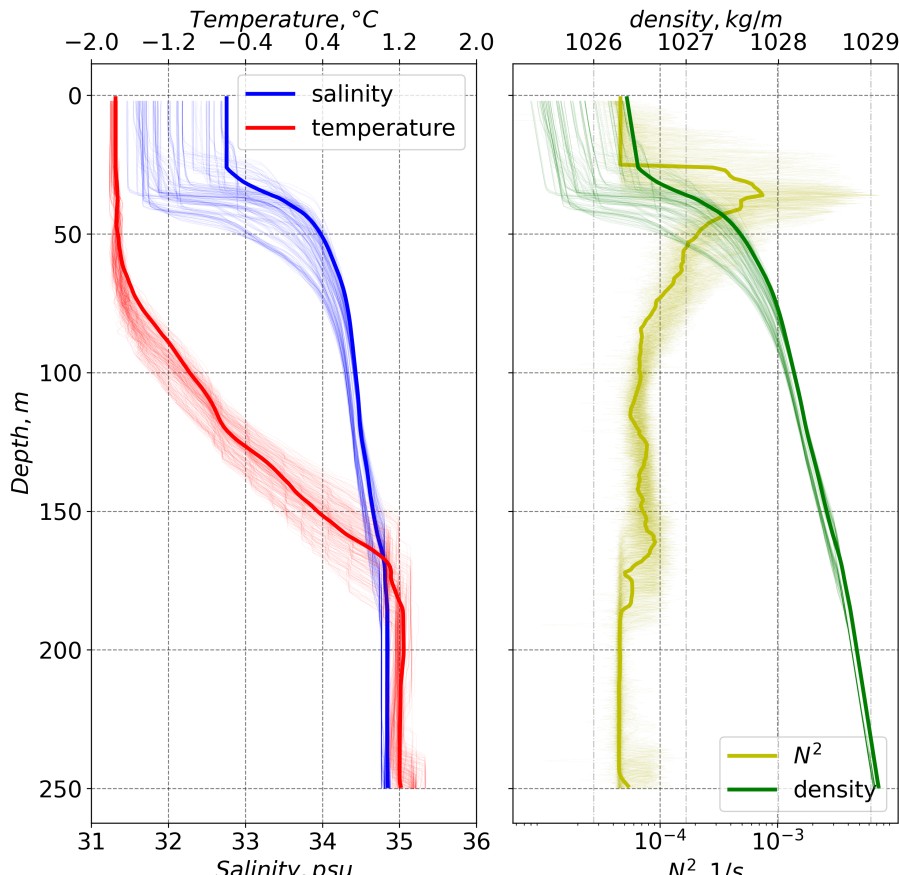

**Figure 4.** Bold lines temperature, salinity, density, and buoyancy frequency ($N^2$) profiles used as initial conditions in the model; thin lines - profiles of the independent MSS data used for the model validation (see Section 3.1).The buoyancy frequency is defined as $N^2 = \frac{g}{\rho}\frac{d\rho}{dz}$, where $g$ is the acceleration of gravity, $\rho$ the density and $z$ the depth.

seen in the north-south section. However, the slope of the isopycnals from west to east is less consistent, below approximately 40 m compared to the north-south section. The same standard deviation in ML depth characterizes both directions. ML depth changes from 27 to 40 m from east to west with a mean gradient of -0.084 m/km and 0.6 m/km gradient standard deviation. In reality, low-salinity intrusions into the ML from the surface can be attributed to changes in both surface heat and salt fluxes. However, in this study, the influence of these fluxes is simulated by nudging, suggesting that the submesoscale variability of the ML depth is most likely governed by eddy dynamics.

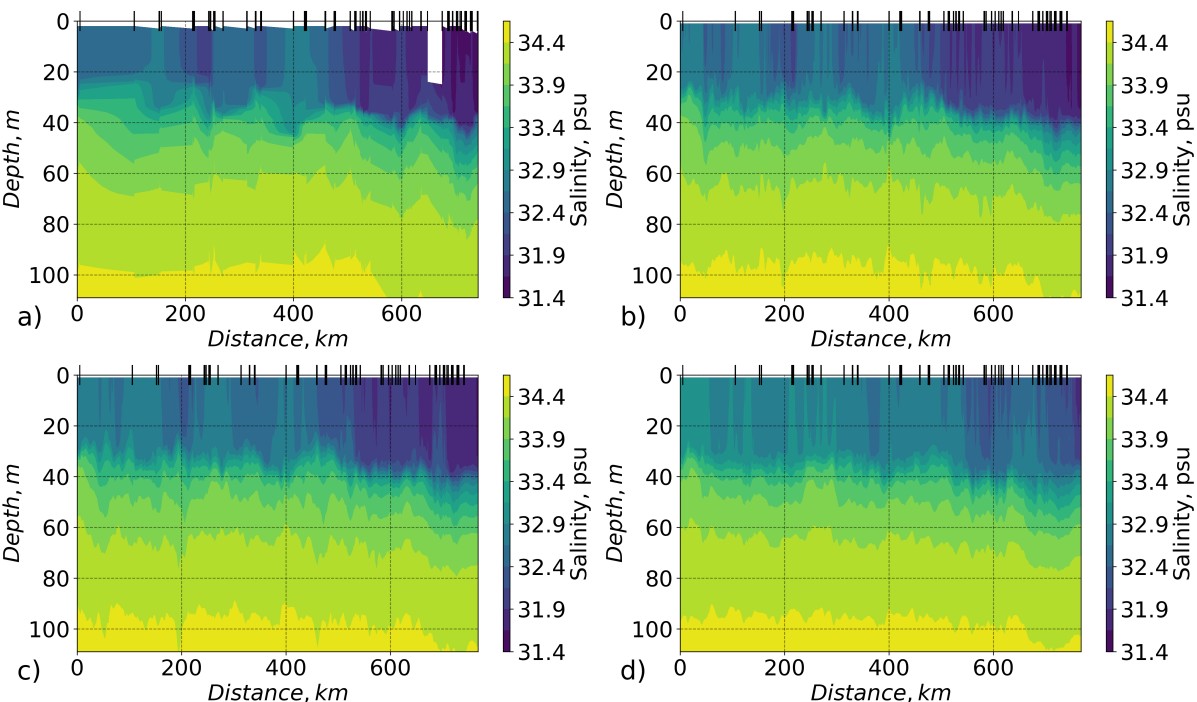

**Figure 5.** Salinity in the upper 120 m along the ship drift path. The x-axis shows the distance of the ship drifts, which is 0 km, at the position of the first MSS profile. Positions of the MSS profiles are marked by black vertical lines at the top axes. a) - salinity measured by the MSS profiler (Schulz et al., 2022) at Ocean City in vicinity of the ship. Here, linear interpolation between MSS casts is applied. b),c) and d) - modeled salinity at the ship positions: b) - simulation with nudging. c) - free run, 2.5 days after nudging was stopped. d) - free run, 19 days after nudging was stopped.

**Table 1.** Root mean square error (RMSE) between measured by SIT buoys and modeled salinity/temperature for simulation with nudging and free run after 2.5 days and standard deviation (SD) of both at different depths.

| depth | RMSE nudging | SD nudging | RMSE free run | SD free run | SD SIT |
|-------|--------------|------------|---------------|-------------|--------|
| 10 | 0.29 / 0.01 | 0.34 / 0.02 | 0.37 / 0.02 | 0.35 / 0.02 | 0.4 / 0.02 |
| 20 | 0.37 / 0.02 | 0.34 / 0.02 | 0.44 / 0.02 | 0.35 / 0.02 | 0.38 / 0.02 |
| 50 | 0.08 / 0.02 | 0.2 / 0.02 | 0.1 / 0.02 | 0.19 / 0.02 | 0.23 / 0.03 |
| 75 | 0.03 / 0.07 | 0.06 / 0.09 | 0.04 / 0.09 | 0.06 / 0.08 | 0.08 / 0.12 |
| 100 | 0.02 / 0.1 | 0.03 / 0.15 | 0.02 / 0.13 | 0.02 / 0.14 | 0.03 / 0.19 |

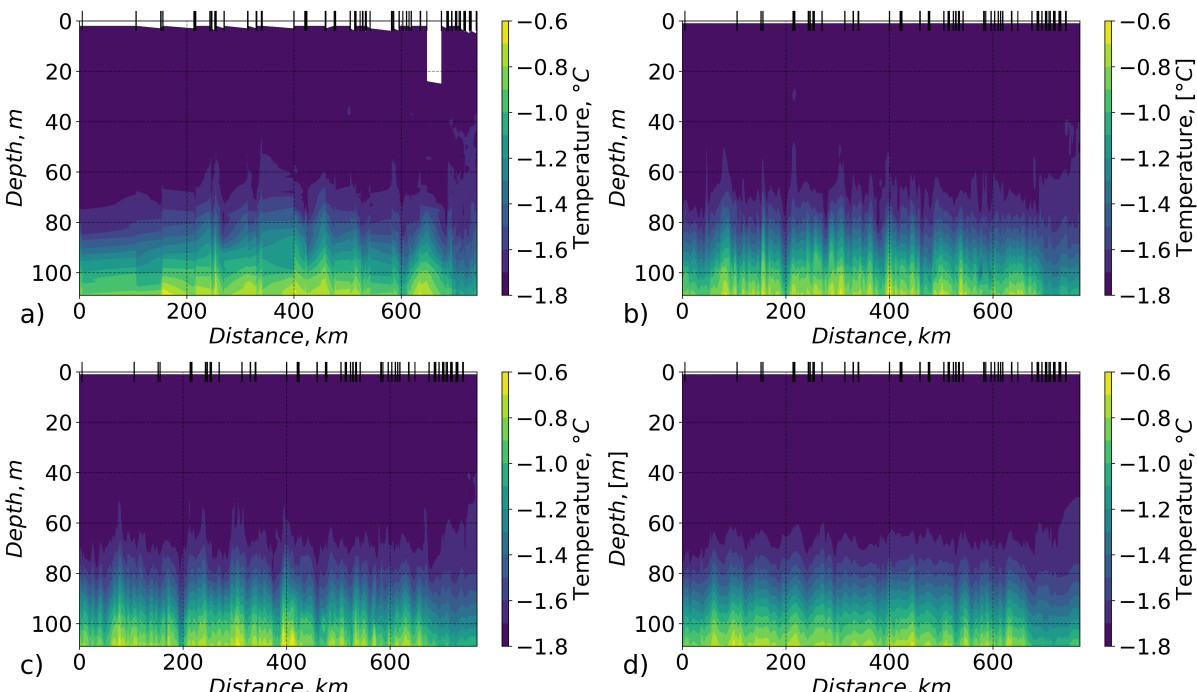

**Figure 6.** Temperature in the upper 120 m along the ship drift path. The x-axis shows the distance of the ship drifts, which is 0 km, at the position of the first MSS profile. Positions of the MSS profiles are marked by black vertical lines at the top axes. a) - temperature measured by the MSS profiler (Schulz et al., 2022) at Ocean City in vicinity of the ship. Here, linear interpolation between MSS casts is applied. b),c) and d) - modeled temperature at the ship positions: b) - simulation with nudging. c) - free run, 2.5 days after nudging was stopped. d) free run, 19 days after nudging was stopped.

## 4 Discussion

### 4.1 Eddy kinetic energy

345 Commonly, modeled eddy kinetic energy (EKE) is defined as the difference between total and mean kinetic energy (Wang et al., 2020). The current setup has no external forcing other than nudging to data. The model does not produce significant mean velocity without external forcing, resulting in negligible mean kinetic energy. Therefore, the total kinetic energy is mainly defined by the anomaly in the velocity and linked to eddies. The EKE is calculated here as

$$EKE = (u^2 + v^2)/2. \tag{5}$$

350 The EKE decreases in the free run with time due to dissipating eddies, as an effect of surface friction or numerical diffusion. The absence of a mechanism to generate new EKE (and new eddies) leads to a decrease in the free run, whereas eddies that formed in the run with nudging remain in the free run for more than 20 days. Figure 9 indicates enhanced modeled EKE activity

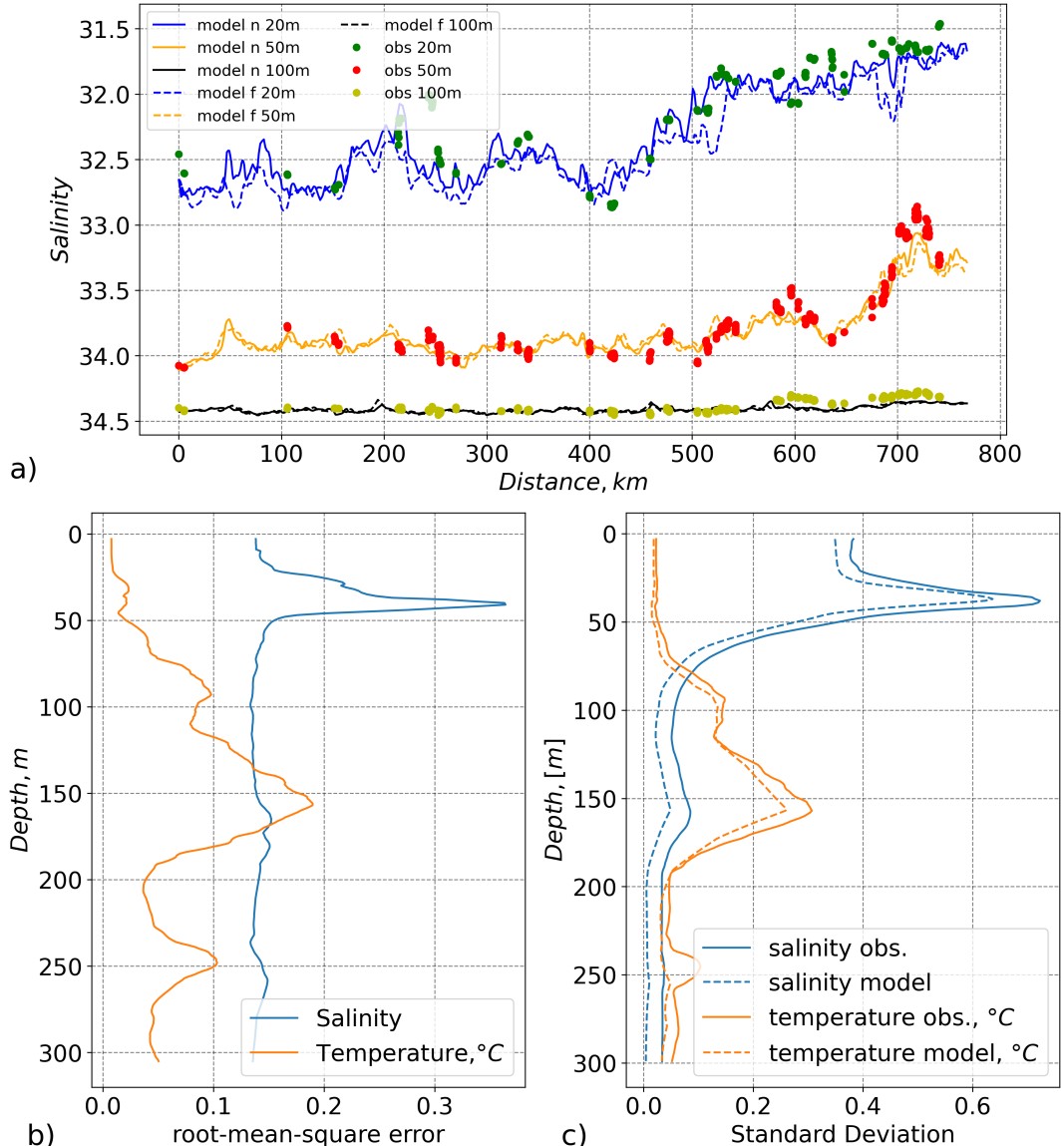

**Figure 7.** a) - salinity at the three depths of 20,50 and 100 m. Y-axis is inverted. Coloured dots - observed salinity data (MSS) from 10 (green), 50 (red), and 100 (orange) m depth. Coloured lines – modeled salinity extracted at ship positions. Solid lines - model with nudging, dashed lines - free run after 2.5 days. b) - root mean square error of model with nudging compared to observational data (MSS). c) - Standard deviation, solid lines - observations, dashed lines - model with nudging.

within two separate depth ranges, both of which are around maxima in $N^2$ (as defined in Figure 4): one in the halocline and the other in the warm Atlantic Water. Similar vertical distributions of EKE in the ice-covered central Arctic basins have been observed previously by Meneghello et al. (2021) and modeled by Wang et al. (2020), and such a bimodal distribution of eddies

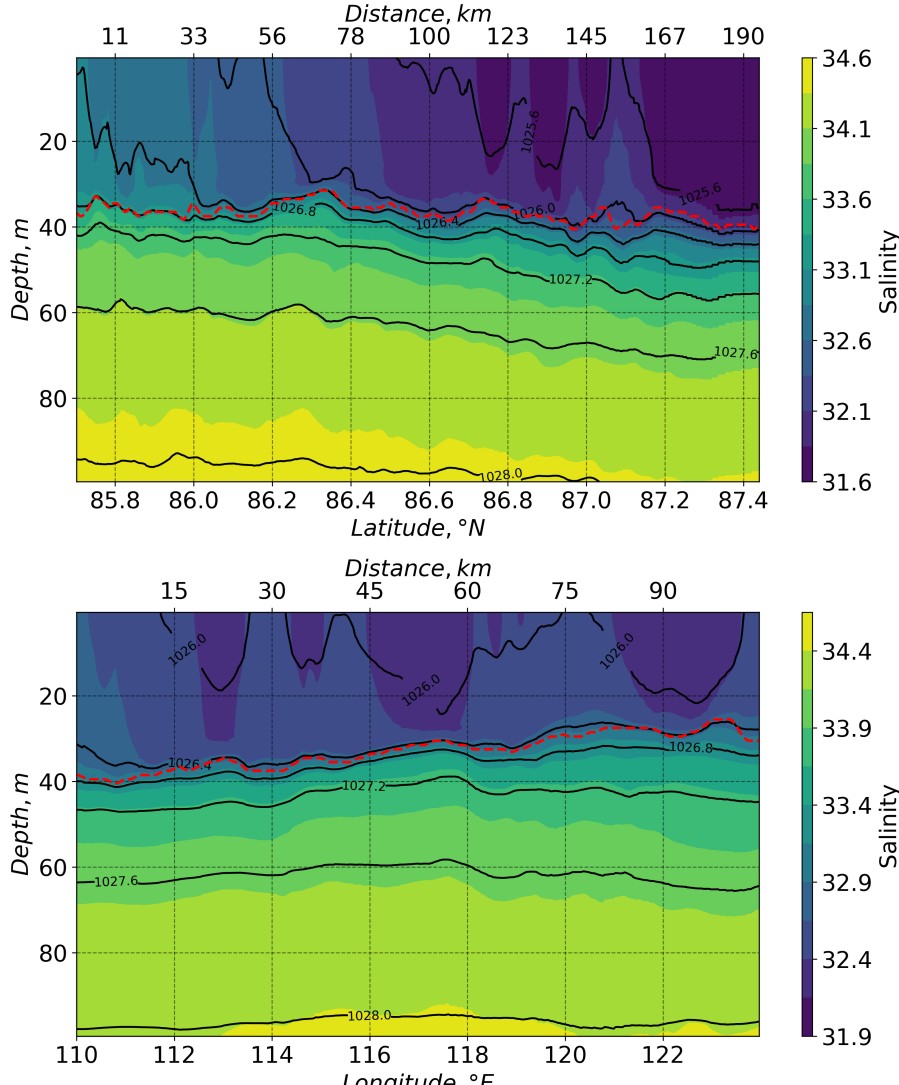

**Figure 8.** Cross sections along 115 °East Longitude (upper) and 86.2 °North Latitude (lower). Filled colour contours are salinity psu, and black contours are density $\mathrm{kg/m^3}$. Red dashed lines show ML depth. Results of the free run after 2.5 days.

was discussed in detail by Zhao et al. (2014). Meneghello et al. (2021) show that subsurface eddies can be explained by the baroclinic stratification and potential vorticity gradients commonly present in the Arctic interior and note the presence of EKE maxima in two layers of maximum density gradients in the Canada basin. However, the data only cover part of the winter, so we cannot analyze the effects of seasonal variability and the associated ice conditions on the distribution of eddies. The ice cover was already formed and consolidated throughout the measurements used here with only a few small openings (Nicolaus et al. (2022)).

We can see a difference in the distribution of our modeled EKE: in the southern part of the domain, the maximum energy was around the warm Atlantic Water, whereas in the northern part, it was intensified in the halocline, just underneath the surface ML. The latter is associated with stronger stratification due to high salinity gradients. In the northern part of the considered domain, there were sharp changes in the direction of the ice drift, which, in turn, also affected the salinity distribution of the near-surface waters. Here, it should be noted that the northern part is covered by data with many overlaps in the drift trajectories (orange polygon in Figure 1 and 9). This, on the one hand, leads to the smoothing of the fields in the model compared to observations where trajectories intersect. On the other hand, it can lead to the appearance of local fronts at submesoscales during nudging. Using a free run partially removes the latter problem.

## 4.2  Eddy examples

As noted, the system achieved a stable numerical solution by the end of the period when the model was nudged towards the data. However, after the external force in the form of nudging is removed, the system begins to change. One can study the dynamics of the formed eddies by examining the changes during the free run. The velocity structure remains similar during 19 days of the free run (see Figure 10), and most of the eddies changing shape and intensity remain close to where they formed. The number of eddies formed in the model during the fairly fast and straight drift is similar to that in the area with overlapping drift trajectories. Most eddies travel much slower than the drift of the ship in the area where the peak in EKE can be seen in the warm Atlantic Water.

The eddy dynamics differ between the northern and southern parts of our region of interest: A few of the small-scale eddies and filaments formed at the depth of the halocline dissipate within a few days in the free run. Among the remaining are those who actively travel and interact with each other. Figure 11 shows an example of a simulated anticyclonic eddy with negative relative vorticity that, during the free run, travels and interacts with a bigger-size cyclonic eddy (positive relative vorticity). The anticyclonic eddy is between 30 and 90 m deep and about 5 km in diameter. The cyclonic eddy is slightly larger and changes its horizontal dimensions from 7 to 10 $\mathrm{km}$. Unlike the cyclonic one with more diffuse boundaries, the anticyclonic eddy has clearly defined contours. The maximum velocities within the eddies reach 12 $\mathrm{cm/s}$. The anticyclonic eddy first travels toward the cyclonic eddy, then circumvents it. The cyclonic eddy remains in position before the anticyclonic eddy approaches it, but once they meet, it begins to stretch towards the anticyclonic one. Changing shape, the cyclonic eddy starts to move north due to the two eddies interacting. The center of the cyclonic eddy moved about 7 $\mathrm{km}$ over 19 $\mathrm{days}$. The anticyclonic eddy starts to spin around the cyclonic one and increases translation speed. On its way, it changes shape from almost a circle to an elongated ellipse and back several times depending on its relative position to the anticyclone eddy.

This example demonstrates that the interpretation of unevenly distributed observational data, sometimes overlapping in space at different times (buoy drift trajectory loops), is complicated but that the drifting buoy observations captured the cyclonic eddy. This is attributed to the quasi-steady nature of the eddy at the time when DN passed through the eddy position. The development of the fast-travelling anticyclonic eddy could not be measured entirely by the DN, as the speed of the eddy was not aligned with the buoy drift at the time, which was northwest. At the same time, the distances between the buoys are larger than the eddy core diameters in our model, which would lead to misinterpretation during the analysis of such data by common interpolation

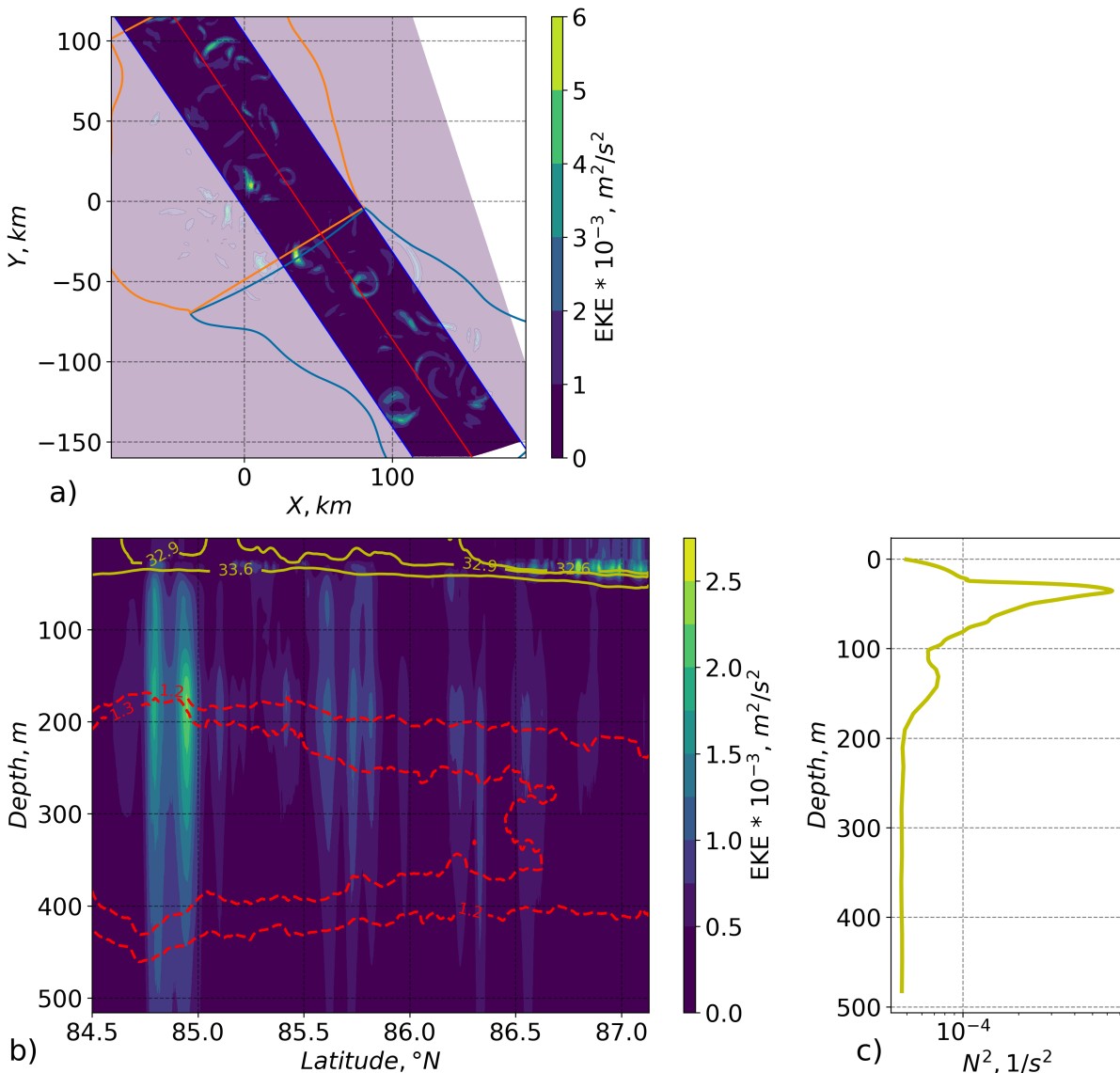

**Figure 9.** Modeled eddy kinetic energy and $N^2$ after 5 days of free run. a) - EKE at 47 m depth. The highlighted area indicates an averaging area in the stereographic projection. The area of averaging is chosen so that the line length along the X direction is the same for any Y position. Averaging was done along X direction and covered 80 km with the center indicated by the red line. The red line was chosen to be the longest straight line within the area covered by data. Green and orange polygons indicate the same areas as in Figure 1. b) - vertical distribution of the averaged along X direction EKE. Red dashed lines are isotherms for 1.2 and 1.3 °C. Yellow solid lines are isohalines for 33.6 and 32.9 psu. c) panel shows mean stratification computed over the area shown in panel a).

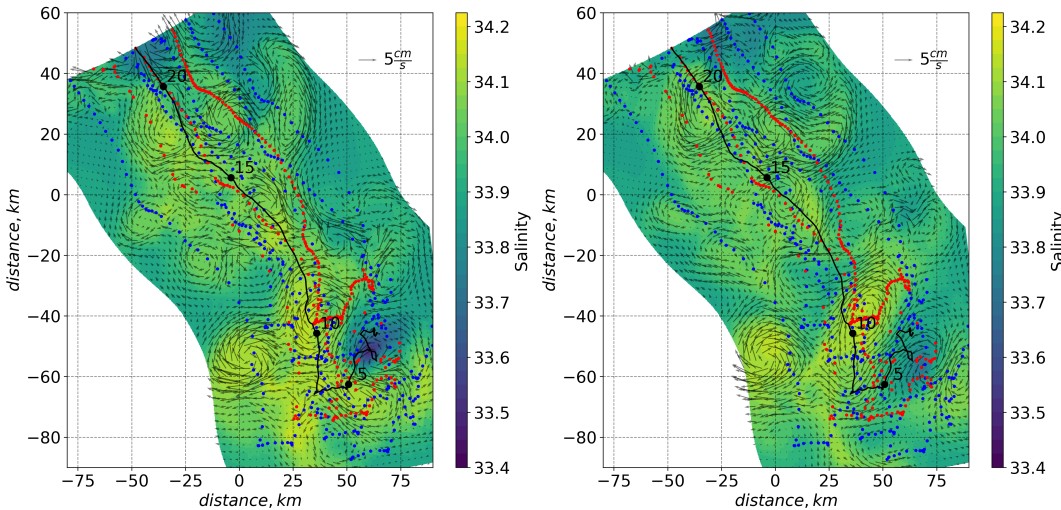

**Figure 10.** Salinity at 47 m depth in the free run after 5 (left) and 19 (right) days model time, with velocity vectors (arrow top right indicate 5 cm/s) zoomed to the area with mainly straight drift (indicated by a green contour in Figure 1). The black line indicates the drift track of *Polarstern*. The black dots are the daily positions of the ship.

methods. This generally applies to the DN measurements, as we only obtained a snapshot in space and time with the DN observing over the scales set by the different DN sites.

The dynamic structures of the fields in the northern and southern parts are different. The nature of the ice drift can explain this. Mahadevan et al. (2010) have shown that submesoscale near-surface eddies dissipate faster under constantly unidirectional 
drift. The ice drift was faster with few changes in direction, unlike the northern part, where direction changed frequently. In addition, horizontal density gradients were observed in the surface layer there, so that near-surface vortices were more likely to form in the northern than the southern part. However, deeper eddies without a dynamical connection to the surface, e.g., within the halocline or the warm Atlantic Water, are not likely to be affected.

### 4.3   Limitations of our method

Changes in the flows due to atmospheric influences are exclusively accounted for through nudging, which is determined by the density of the data and shows a smoothed pattern. In the quasi-stationary case, considering that the observational data have a temporal spread of several months, this can result in horizontal gradients in temperature and salinity fields. For example, we can expect a decrease in temperature and an increase in salinity in the mixed layer due to ice formation in the temporal range from November to January. Considering that the buoy drift was northwest during this time, with such a quasi-stationary approach, 
we can expect an increase in salinity and density in the north and west directions of the mixed layer. However, according to the simulation results, we observe the opposite pattern: no gradient in the west-east direction and a reverse gradient from north to

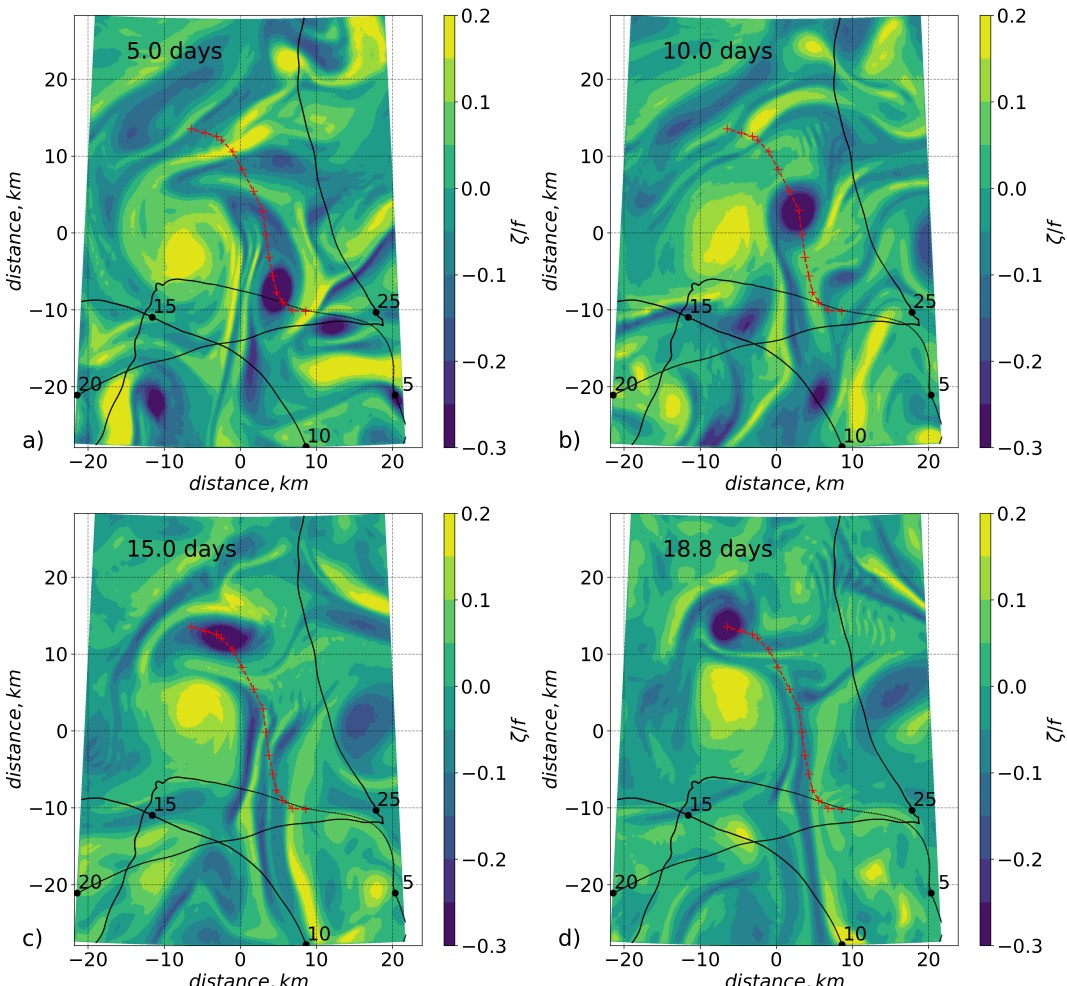

**Figure 11.** Snapshots of relative vorticity at 47 m depth in the free run. The ML depth is about 27 m. a),b),d), and c) - snapshots after 5,10,15, and 18.8 days of model time, respectively. The black line indicates the drift track of *Polarstern*. The black dots represent the daily positions of the ship, starting from 2019-11-12. The red dashed line shows the trajectory of the center of the anticyclonic eddy described in the text, where the center was identified by eye. The position of the region shown in this figure is denoted by the magenta box in Figure 1.

south (see Section 3.2). Thus, it can be assumed that the instantaneous gradient (a gradient that could be measured at a single point in time) in the north-south direction is more significant than in the reconstruction.

Despite significantly reducing the influence of regional boundaries on the final solution through increased model resolution,
there remains an "internal boundary" in the model between regions with available data and regions without observations. This creates an artificial front between these two areas, determined by the initial conditions. Instabilities can occur along the boundary of this front due to the dynamics in the data-rich area and the absence of dynamics in the data-sparse area. This issue

is partially mitigated by conducting a long spin-up calculation on a coarse grid, which significantly smooths the front between the areas.

In our simulations, there are no barotropic currents or currents caused by baroclinic gradients on the scales of the Arctic basins. Typical time-averaged velocities for such currents can be 2-5 cm/s (Rudels, 2009). In situ velocity measurements during the MOSAiC experiment show average velocity values of about 3 cm/s at depths of 60-200 m (Figure 11 in Rabe et al., 2022; Baumann et al., 2021). The average velocity values across the entire area for the same depth in our simulations range from 1-2 cm/s depending on the depth, with peaks up to 15 cm/s in eddies. Thus, we can assume that the influence of basin-scale

dynamics not considered in our work has a relatively minor effect on the final solution. In future work, it would be worthwhile to use ocean and atmosphere reanalysis data (which utilize MOSAiC data) to provide initial and boundary conditions if and when they become available. Despite these reanalyses' coarse vertical and horizontal resolutions, their usage would allow transitioning from a quasi-stationary case to a time-dependent solution.

## 5    Summary

The MOSAiC project has collected a rich data set of physical oceanography observations in the central Arctic. Various measurement techniques and tools make combining and analyzing the obtained data challenging. This paper presents ocean model reanalyses of the first part of the MOSAiC field campaign during winter. Three-dimensional temperature and salinity fields were reconstructed by model nudging to the observed data. The model setup covers the Arctic region bounded by 84.5°to 87.5°North and 87.6°to 139.5°East, corresponding to the MOSAiC drift from October 2019 to January 2020. Using the re-

gional FESOM-C model has allowed us to analyze the dynamic fields.

The model was further developed to suit our needs. The turbulence closure was adapted specifically for this work by modification of the turbulence scale. Further, we extended the model with a semi-implicit method to calculate the sea level. The model code was parallelized with MPI libraries, which made it possible to perform calculations with the required resolution and 1.3 million horizontal nodes. Considering 240 vertical layers for the current setup, the number of calculated model points is

more than three times that of the eddy-permitting global mesh with a quasi-uniform resolution of 15 km in the FESOM2 setup (Danilov et al., 2017). Our developed setup of the FESOM-C model with an unstructured mesh makes it possible to mitigate the influence of boundary conditions on the area of interest. Near-surface and deep submesoscale processes are resolved by the high horizontal resolution of up to 250 $m$ and the vertical resolution of up to 1 $m$. A simple algorithm was developed to nudge the model to the observed data, which makes it possible to use a large amount of data from different measuring systems. More

than 630,000 single-point temperature and salinity observations and over a thousand vertical profiles were used to nudge the model.

We validate the model output against independent data that were not used for nudging, and the model well reproduces the model's vertical and horizontal distributions of temperature and salinity. The main discrepancies between independent and modeled data are at the intersections of the drift trajectories of the buoys. Buoys have overlapping tracks due to the ice drift

loops, for example, during rapid changes in the direction of the wind. These not-quite-synoptic measurements lead to a gridded

(nudging) field that is not necessarily equal to individual observed values due to the weighted mean interpolation. These crossing points could be utilized as temporal references to calculate errors associated with the 'quasi-stationary' assumption in future publications concerning these data and the model.

We have reconstructed dynamically consistent three-dimensional temperature, salinity, and density fields by employing the model nudging to data method and a model-free run. Our simulation results allow for the analysis of the horizontal and vertical distribution of temperature, salinity, and velocity on a regular grid. Our analysis of the dynamic characteristics reveals the existence of two separate depth ranges of enhanced eddy kinetic energy located around two maxima in buoyancy frequency in the central Arctic basins. This bimodal distribution of eddies, previously noted in various studies, shows regional variations: in the southern domain, maximum energy is near the warm Atlantic Water, while in the northern part, it's intensified in the halocline beneath the surface mixed layer. The model resolves several stationary warm-Atlantic-Water eddies and provides insights into the associated dynamics that would not be possible by analyzing the observations alone.

This study presents potential for further research and practical applications. The reconstructed physical fields can serve as a foundation for analyzing the dynamics of the halocline and the transport of organic and inorganic matter within the water column. Furthermore, the dynamic fields we have generated offer a valuable tool for assessing the impact of submesoscale dynamics in the Arctic during winter on vertical exchange processes. These insights can also inform the planning of expeditions and the deployment of autonomous buoys.

For future development, coupling existing simulations with online or offline biogeochemical models could provide an understanding of the Arctic's winter ecosystem. Another realistic development is extending this method to non-stationary solutions that could use global reanalysis data for both the ocean and atmosphere, coupled with more realistic ice parameterizations. These developments, combined with other data from the ice and atmosphere collected during the MOSAiC expedition, could advance our ability to investigate the evolution of submesoscale dynamics.

*Code and data availability.* The FESOM-C code and model setup used in this study can be found under DOI:https://doi.org/10.5281/zenodo.8004904. Modelling results for 6 vertical layers and 5 temporal layers of the free run created during this study are openly available in the Zenodo open data repository and can also be found at the provided link. The rest of the modelling data are available upon request by contacting the corresponding authors.

*Author contributions.* IK, BR, YCF and AA were responsible for the initial conceptualization of the study. IK developed the model nudging method, model setup, simulations, and processing the results and wrote the paper with a strong contribution from BR and AA. BR, YCF, MH, IK, AQZ, ST, KS, VM, IF, MJ contributed with providing, post-processing and curation the observational data, and deploying instruments in the field. AA, IK, SH, VF developed the FESOM-C code. All co-authors reviewed the manuscript and contributed to the writing and final editing.

*Competing interests.* Ilker Fer is a member of the editorial board of Ocean Science.

*Acknowledgements.* Data presented in this paper were produced as part of the international Multidisciplinary drifting Observatory for the Study of the Arctic Climate (MOSAiC) with the tag MOSAiC20192020 (grant numbers AWI_PS122_00 and AFMOSAiC-1_00). The instruments were funded by the MIDO (Multidisciplinary Ice-based Drifting Observatory) infrastructure. The study contributes to the Changing

485  Arctic Ocean (CAO) program, jointly funded by the UKRI Natural Environment Research Council (NERC) and the BMBF, project Advective Pathways of nutrients and key Ecological substances in the ARctic (APEAR) grants NE/R012865/1, NE/R012865/2 and #03V01461; the project EPICA in the research theme MARE:N - Polarforschung/MOSAiC funded by the German Federal Ministry for Education and Research with funding number 03F0889A; the European Commission for EU H2020 grant no. 101003472 (project Arctic PASSION), and the AROMA (Arctic Ocean mixing processes and vertical fluxes of energy and matter) project by the Research Council of Norway, grant

490  number 294396. We thank the administrators of the AWI cluster Ollie and Albedo, where the simulations took place, for their continual support and patience. Full acknowledgements are available in Nixdorf et al. (2021).

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
