# Peer review of "Dynamical reconstruction of the upper-ocean state in the Central Arctic during the winter period of the MOSAiC Expedition"

_EGUsphere, 2023_

## Author Comment (AC1)

**Response to Reviewers for Paper:** "Dynamical reconstruction of the upper-ocean state in the Central Arctic during the winter period of the MOSAiC Expedition."

**Reviewer 1:** *The article presents a model reconstruction of the Arctic Ocean structure during the winter period of the MOSAiC Expedition. The authors used the FESOM2 model with an altered turbulence closure scheme at high resolution. The model results were nudged using profile measurements from buoys, and evaluated against an independent set of profile measurements. The resulting model simulation shows signs of enhanced eddy kinetic energy around the halocline and the depth of the warm Atlantic Water.*

*While the method seems by-and-large reasonable, in my opinion additional work needs to be done in the analysis and the description of the work in order for it to be ready for publication. A few general comments:*

*The authors should clarify the language used throughout. While the methods describing nudging the model using the data, the phrase "nudging of the data" is frequently used, implying that the data where being altered by the model. This should be clarified.*

**Response:** The authors are grateful for the reviewer's overall feedback and find it highly valuable. The specific points raised have been addressed in the following sections. A small remark: we are developing and using the FESOM-C model, not the FESOM2 model. Although these models are from the same family, their differences are described in the article.

The phrases have been changed to "model nudging" or "model nudging to data" throughout the text.

**Reviewer 1:** *In its current form, the introduction reads like a list of relevant papers. The paper would be strengthened by integrating the results of prior work into a description of the state of the science for the relevant processes, instead.*

**Response:** We have revised the introduction to provide a better flow, leading up to the objectives of the manuscript. Selected citations have been added or removed.

**Reviewer 1:** *The authors find that there are discontinuities introduced by locations where the trajectories of the buoys form loops. In my mind, this indicates that the ocean is evolving and that treating the observations as a frozen-in-time snapshot is a problem. Perhaps it makes sense at certain time scales and for certain depths.*

**Response:** We concur with the reviewer that duplicate data introduces uncertainties in data interpretation. For instance, it is unrealistic to expect congruence in measurements taken at different phases in the context of inertial waves. This is one of the reasons we implemented a 1 km distance in our nudging method (refer to line 228) within the "loops" region where the model approximates the data. With this approach, especially in areas where trajectories intersect and the data resolution is high, the values for nudging are smoothed, preventing the formation or disruption of a baroclinic front. This also enhances the stability of the model's solution.

In scenarios where ocean dynamics undergo significant changes (relative to

ice drift) within a region containing rapidly moving eddies, we address this in our example concerning eddies movement in a model without nudging. In such cases, data interpretation necessitates modeling without nudging, thereby rejecting the quasi-stationary approximation assumption. However, as illustrated in Figure 5, a free run does not substantially alter the mean characteristics of the reconstructed fields. Generally, intersections of buoy trajectories are brief, spanning short durations relative to the several months over which measurements are conducted.

**Reviewer 1:** *The colormaps used in Figure 9 and 7 should be replaced with colorblind-friendly and print-safe colors.*

**Response:** The colormaps of Figures 9 and 7 are changed in the revised manuscript.

**Reviewer 1:** *Minor grammar and typography errors throughout, some are listed below.*

**Response:** All minor comments are addressed in the revised manuscript.

**Reviewer 1:** *More importantly, it's not clear to me what the key contribution of the paper is. This is not to say the work isn't valuable or worthy of publication. Rather, I think that substantial revision of the introduction, discussion, and summary is needed to clarify the importance of the work. Clearly a lot of thought and effort have gone into this, and I think restructuring the presentation can bring the value of the work more clearly into focus.*

**Response:** The introduction is rewritten in the revised manuscript. Additionally, the discussion and summary sections are modified to emphasize the main messages of the article regarding the potential application of nudging for reconstructing three-dimensional fields in this experiment, with implications for their use in other research. This also includes a demonstration of the distribution of Eddy Kinetic Energy (EKE) in the central Arctic region.

**Reviewer 1:** *A few (non-exhaustive) minor comments:*

*6 "drift speed direction" = "drift speed and direction"?*

corrected

*13 "And no" – "Simulations show no. . . " or something like that?*

corrected

*18 capitalization unnecessary for "earth system models"*

corrected

*21 Grammar unclear*

corrected

*31 Grammar*

corrected

*75 "so-called" implies that there is some doubt in the name. The site is called Ocean City*

corrected

*80 Define "DN"*

corrected

*85 "one possible method are" grammer incorrect, could replace with e.g. "one possible approach is to use interpolation techniques"*

corrected

*191 DN buoys trajectories → "DN buoy trajectories" or "trajectories of the DN buoys"*

corrected

*193-4 "these interlacement" unusual word choice, I'd rephrase for clarity*

corrected

*195-196 – Why would we expect the measurements to be the same after a repeat visit? I don't understand why this would lead to aliasing of a signal.*

Aliasing of the signal occurs only in a quasi-stationary approximation during spatial analysis, such as in the case of horizontal interpolation. The phrase has been removed to prevent confusion.

*203 (and throughout, including in the summary). "Nudging of the data" implies that you are altering the data. Is it not the case that you are nudging the model using the data?*

Certainly, the model is nudged to the data. Changed throughout the text. Confusion due to "data assimilation".

*400 – What is meant by "or October 2019 to January 2020" here?*

From October to January, the DN drifted within these geographical boundaries. The sentence has been reformulated.

---

## Author Comment (AC2)

**Response to Reviewers for Paper:** "Dynamical reconstruction of the upper-ocean state in the Central Arctic during the winter period of the MOSAiC Expedition."

**Reviewer 2:** *A review of "Dynamical reconstruction of the upper-ocean state in the Central Arctic during the winter period of the MOSAiC Expedition" by Kuznetsov and co-authors. My background in is modeling and data assimilation of the Arctic Ocean, but not so much sub-mesoscale oceanography.*

*The manuscript exploits a very dense measurement campaign from the ambitious MOSAiC ice camp somewhere in the central Arctic and assimilates it into a numerical model of very high resolution. As there is no other realistic ocean model of similar resolution set up in the central Arctic to my knowledge, the study stands out by its very high originality.*

*The closest relatives of such studies were the pioneering ocean forecasts by the Harvard Group in the early 90's where cruise data were assimilated by Optimal Interpolation and fed into a forecast model running onboard during the cruise (Robinson et al. 1996), which did demonstrate forecasting skills. The MOSAiC was however on a slower path at the speed of the sea ice drift and was not in control of the trajectory, so the ambitions stay realistically on a lower level, that of performing an oceanographic process study, which is interesting in its own right.*

*The authors did interpolate the temperature and salinity profiles obtained along the whole 4-months experimental period into 3D fields and assimilated these data into a bespoke model set up specifically for the area of the experiment, and then discuss the dynamical features of the simulated ocean fields.*

**Response:** Thank you very much to the reviewer for such detailed and constructive comments. We want to avoid misunderstandings from the outset and clarify the method we used. The observed data were indeed extended onto the model grid to further fit the model to this data (lines 217-219 of the original manuscript). However, it occurred only at grid nodes no more than 1 km from the observation point. The number of grid nodes constrained the influence; with a horizontal resolution of 1 km, the observational data impacted no more than 3 model nodes. Conversely, with a resolution of 0.25 km, the data's effect was confined to the 30 nearest nodes and spanned less than 1 km. In other words, with a distance of 20 km between the buoys, the zone not covered by the data spanned 19 km. Moreover, there was no vertical interpolation as such since observational data with a resolution of 1 m were used for data from autonomous profiles. For other data, from the CTD chain buoys, the data from the observation horizons were expanded vertically by 3 m. In both cases, the expansion of the zone of influence of the data on the model (1 km horizontally and 3 m vertically) was implemented to avoid the formation of sharp horizontal fronts when crossing buoy trajectories and to prevent strong vertical instabilities when crossing trajectories of different types of instruments, which could lead to an unstable model solution. This approach is not about interpolating data and then using these 3D interpolated fields to nudge the model, as the nudging was performed in relatively limited spatial areas.

**Reviewer 2:** *A major assumption of the study is therefore the synoptic*

*nature of the measurements assimilated ("quasi-steady-state" is not mentioned until line 203 in the methods description, which is very late for an important assumption). Further on, the model results reveal that this assumption was actually wrong, as acknowledged in the conclusion. This is a major weakness of the study that should be recognised from the start to avoid some unnecessary suspense.*

**Response:** Application of the quasi-steady-state assumption is now included in the abstract in the revised manuscript. Additionally, more justification for the application is added to the methods section.

The absence of full 3D interpolation (interpolation over the entire space between the data, as mentioned above) provided us with the opportunity to apply the quasi-steady-state assumption. This is precisely because the distance between the buoys and observation points is greater than the characteristic size of submesoscale eddies (7 km) in this area. Specifically, because there is no horizontal interpolation between the data, we use this approximation, assuming that sub-mesoscale processes do not influence each other on mesoscale scales (50 km). The example we provide of the interaction of eddies clearly demonstrates that using any horizontal interpolation of temporally separated data can lead to challenges in identifying submesoscale processes. It should also be noted that the dynamics shown in the latter part of the article, which the reviewer refers to, were simulated in a free run when the data did not constrain the model, and the quasi-steady-state assumption was not applied.

**Reviewer 2** *Another major weakness is the odd choice of the interpolation method. The inverse distance method is not able to de-cluster observations in an irregular network such as the ones obtained here during a nearly random drift, and generates very erratic extrapolation features, so it should not be used other than with regularly spaced measurements. It is possible that the inverse distance worked well when combined with a clustering method briefly referred to, but it is a priori an ill-informed choice (See Zimmermann et al. 1999 for a thorough comparison, as well as modern examples of interpolation with Bourgain and Gascard 2011 and Troupin et al. 2012).*

**Response:** As we have already noted, there was no 3D interpolation for all nodes of the model. We used the method described in the technique solely to smooth the effect of high-resolution data and to smooth data from different sources; this is more like extrapolation.

**Reviewer 2**: *I will argue later that the study may have been improved by extracting only the large scales variations from the data and perturbing randomly the mesoscales.*

**Response:** The experiment with random perturbations is interesting as a sensitivity analysis of the model. Similar experiments were conducted in the earlier stages of developing mathematical modeling methods, including the FESOM model family. It should also be noted that, due to the absence of continuous interpolation across the entire area, we are implementing in this paper considering only that the perturbations are not random but are based on data.

**Reviewer 2**: *The resulting maps of interpolated values, the only energy*

*source of the simulation are not shown, which casts the shadow of a doubt on the realism of all the results obtained throughout the paper. Are we looking at observed mesoscale features or quasi-random (sampling-dependent) perturbations of a homogeneous density field?*

**Response:** Figures 5, 6, and 8 show the resulting maps and transects, reconstructed using the model. We do not have interpolation maps as noted above. We have a 'spaghetti plot' from the data with an increased nudging radius of up to 1 km. An example can be seen in the article https://doi.org/10.5194/essd-14-4901-2022, Figure 7a.

**Reviewer 2:** *The validation against independent observations is unconvincing because these measurements are taken in the vicinity of the ice camp and are not representative of the remote unobserved areas. The authors correctly recognise that the validation is poor at the cross-over points but still boast uncritically the success of the validation in several places. This is not a major problem since the model is used for process studies which do not require any accuracy but the text gives a misleading impression of accuracy.*

**Response:** Independent observational data (near daily temperature and salinity profiles from a microstructure profiler, MSS) were obtained 20 km from the buoys, based on which we adjusted the model. Indeed, they are located at the center of the pattern, but, as previously noted, we do not employ any data interpolation at such scales. The model is not simply superimposed onto the fields. The data from the ship's CTD (located in the vicinity of MSS) used for nudging were primarily collected on different days from when the MSS measurements occurred. Moreover, CTD profiles measured from the ship were conducted once a week, while microstructure profiles were taken almost every day. The data is independent: it is distinct from the rest and was not collected concurrently with the ship's measurements.

**Reviewer 2:** *The authors do not use any numerical model reanalysis nor climatology as background values, which is probably for the best to avoid additional artefacts.*

*There are other aspects of the experimental setup that are should be clearly explained upfront in the paper rather than admitted too late in the discussion section. One is that the ocean is completely shielded from the atmosphere by an idealised ice cover, so that the only source of momentum in the model is the nudging to temperature and salinity.*

**Response:** We discuss ice in the model description lines 125-129. The influence of ice in the model occurs through the parameterization of friction in the surface layer and the parameterization of turbulent exchange coefficients in the closure equation. The suggestion that the ice drift does not serve as a source of momentum due to ice dynamics is added to the model description in the revised manuscript. The idealized ice cover is now mentioned in the abstract.

**Reviewer 2:** *Another one is the breach of the continuity equation by the nudging, which contradicts the assertion that the assimilation is physically consistent.*

**Response:** In the methods section, we've included a note stating that nudging violates the principle of continuity. It's essential to emphasize, however, that our application of data nudging is confined to specific observational points, rather than uniformly across the entire region. This targeted approach avoids major complications while establishing initial conditions for a free simulation phase.

**Reviewer 2:** *The data assimilation method itself is admittedly very rudimentary (nudging), but contains unexpected complications that are not justified at all: using different relaxation times for temperature and salinity and the odd-looking vertical relaxation coefficient in Eq (4). If these complications were necessary then the authors should explain what led to them.*

**Response:** 1. We use the same coefficients for temperature and salinity. Perhaps the reviewer referring to the Ck coefficient ? However, the index 'k' pertains exclusively to the type of instrument, as detailed in the text, and not to a differentiation between temperature and salinity. 2. The 'odd-looking' vertical relaxation coefficient is applied to the first group of data obtained from buoys that record temperature/salinity at five levels. This is to ensure that the model (in the absence of 3D interpolated fields, and with only point data available) does not experience a shock from nudging at a single point. We employed the method described in the article to smooth the effect of nudging. The text added to the article states: Thus, the model's nudging occurs in the vicinity of the observation point +-3 meters, but the strength of the nudging decreases with distance from the observation point.

**Reviewer 2:** *The paper writing is overall quite poor, even though the English is good, the explanations and justifications are often vague and the logic is not obvious. This is particularly true of the introduction, which reads as a long enumeration of unconnected facts. So the paper needs a thorough revision of the text to remove all the loose ends and strengthen the logic.*

**Response:** We have revised the introduction to provide a better flow, leading up to the objectives of the manuscript. Selected citations have been added or removed.

*Before the paper is acceptable for publication, the authors should provide visual evidence that the interpolated fields obtained by the inverse distance method are making sense as a quasi-steady-state estimate of the water masses or if any random perturbation of the homogeneous initial fields would have led to the same conclusions.*

**Response:** As noted in the manuscript on lines 217-218, the nudging term was incorporated solely for grid nodes located in close proximity to observation points. Thus, we do not have interpolated fields obtained by the inverse distance method; we only have scattered data + interpolated/extrapolated values within a radius of 1 km from a specific measurement. The effect of these for nudging was expanded by forming a matrix indicating the presence or absence of data within a 1 km radius of observation points. However, data from a neighbouring buoy located 20 km away does not affect nudging in the vicinity of 1 km from another buoy. The exception is moments when the trajectories of the buoys intersect in space but not in time. In such cases, the method used in the vicinity of 1 km from the observations is practically irrelevant; the main task here is exclusively

to smooth out potential pseudo-fronts caused, for example, by different phases of the internal wave. Random perturbations of the observed data will not lead us to the same results.

**Reviewer 2:** *The introduction should be completely re-written to prepare the reader for the experiments at hand and formulate more precise goals than to "extend current knowledge of submesoscale dynamics". The conclusions are just as vague: they are mostly reflecting a posteriori on the limitations of the experiments rather than highlight the newly gained insights related to the vertical EKE profiles.*

*The paper has important scientific merits in spite of the abundant flow of criticism coming below, so I believe that it should appear after major revisions: new experiments would be an improvement but are not compulsory. However there should be a restructuring of the text, better explanations and a new figure showing horizontal interpolated Temperature and Salinity fields.*

**Response:** The manuscript is rewritten in accordance with the reviewer's suggestions.

**Reviewer 2:** *Detailed comments: The abstract does not work as an abstract because it lacks most of the basic elements of context (What? Where? When? How?). On the contrary the five first lines do not belong in an abstract, but more in the introduction, and can be safely removed.*

*L8: The model is a major element of the study. The reader needs to know what kind of model is used: its nature (ocean general circulation without active sea ice), its name, the mesoscale-resolving resolution.*

*The time period of the study is missing, at least the season would be useful to know.*

*L12: Indications like East and West make no sense unless you mention the name of the area: the Nansen Basin, Amundsen Basin?*

*L12: 'high variability' is also blue sky to the reader. High with respect to what?*

*L16 'the fields can be used for further analysis'. That statement is very vague and should be made more specific once we have an impression of the degree of realism of the interpolated data.*

**Response:** The abstract is now rewritten, taking into account the reviewer's comments.

**Reviewer 2:** *The introduction is an accumulation of facts taken from the literature. Although all of them are interesting in their own right, they cover a too broad scope to frame sufficiently the scientific context of the present study. They also read like an itemised notes from a literature review with no indication whether the findings will be revised by this study or not, and most of them are not. The logical succession of these facts is also left to the imagination of the reader.*

*- L.50: Typically "An analysis of the dynamics of baroclinic vortices [...] is given in Sokolovskiy and Verron (2013)" does not tell whether this analysis is in any way related to the present paper. If the discussion does not loop back to it, then please remove it from the introduction.*

*- L. 68: "Very high horizontal resolution" is too vague. Are they eddy-resolving, permitting, or event in the non-hydrostatic assumption?*

*- L. 85: there are more than one interpolation technique. Since this part is criticising interpolation techniques, it is the adequate place to mention the one that will be used in this paper.*

*- L89 to 97 the whole paragraph is a very cumbersome justification for using rudimentary rather than advanced data assimilation. If we trust your argument as it stands, there is no advantage to advanced data assimilation methods at all (nudging is more practical and yields better results) and nobody should ever be using anything else than nudging. Obviously you do not need to upset the whole data assimilation community to justify your choice of method. It is sufficient to state that 1) the costs and the complexity are not affordable in your case, plus 2) that the data coverage by a single quasi-random track is very unusual, so you lack evidence that advanced data assimilation is cost-effective in your case. Please rewrite the paragraph to better justify the choice of nudging.*

*- L98: The goal of the study "extend current knowledge of submesoscale dynamics" is too vague. It is impossible to verify whether this goal has been attained or not. Please make it more precise.*

**Response:** The introduction is now rewritten, taking into account the reviewer's comments.

**Reviewer 2:** *- L115: Indicate already here the vertical coordinate of the model is sigma rather than in Section 2.3.*

*- L125: There is no thermodynamical effect of sea ice on the ocean, the next section will indicate that the ice drift is a constant value. A missing piece of information here is the sea ice area coverage, which seems to be 100% thus sheltering completely the ocean from the atmosphere. It should be made clear that there is no direct effect of the atmosphere on the ocean and that, after mentioning the constant lateral boundary conditions, there is no input of momentum to the model apart from the nudging term. As recognised somewhere far down in the manuscript.*

**Response:** In the model description, specifically lines 125-129, we address the topic of ice. It is clarified that in our model, ice drift does not act as a source of momentum; instead, its impact is limited to friction and serving as an upper boundary in turbulence closure. This clarification is now included in the model description. The influence of ice in the model occurs through the parameterization of friction in the surface layer and the parameterization of turbulent exchange coefficients in the closure equation.

**Reviewer 2:** *- L139: Is the value of 0.7 m/s set for the whole period and the whole model domain? Please explain why you have not made it more realistic.*

**Response:** This parameter is used across the entire domain and throughout the entire period of the model's nudging. This value describes the upper limit in our parameterization of vertical turbulence and sufficiently well represents turbulent mixing for our task. Since we use a 'quasi-steady-state' approximation, this parameter remains unchanged. More sophisticated parameterizations of ice dynamics and thermodynamics might lead to a more realistic description of the turbulent layer beneath the ice. However, we lack information about the ice

roughness at the water-ice interface, realistic small-scale wind conditions, snow distribution, and many other factors that determine the upper boundary.

**Reviewer 2:** *- L141: Why do you choose this definition of the mixed layer depth and why a minimum of 20 meters?*

**Response:** Because according to our observations, we did not see a mixed layer thinner than 20 meters, the specific number is not important for the task definition, but it simplifies and speeds up the model. Delving into the numerical methods of the specific task goes beyond the scope of this article. This is one of the commonly used definitions of MLD, which does not play a crucial role for the specific task.

**Reviewer 2:** *- L159: Please indicate here the nature of the model boundary conditions. Not later.*

**Response:** The nature of the boundary conditions is described above; they are solid boundaries. If what is meant are the boundaries between zones where there are mostly data and where there are none (line 380), then this is the result of our experiments and is not related to the model setup.

**Reviewer 2:** *- L165: A boarder situation map with some topographic features would help understanding where we are. And where are the North and the East.*

**Response:** A map of the experiment region is now added to Figure 1.

**Reviewer 2:** *- Figure 1a) is too small to discern all the details. I cannot see the cyan rectangle, maybe because I am colour-blind, but I suspect there is too much information on this sub-plot.*

**Response:** The cyan color is now changed.

**Reviewer 2:** *- Figure 1b) shows a wide spread of T/S profiles, but only one density profile, which leaves us to imagine what the spread entails in terms of density changes. Can you include the spread of density still keeping the clarity of the plot.*

**Response:** Changes have been made to Figure 1, including the map of the region, spreads of density and n2.

**Reviewer 2:** *- L190: The duration of the experiment, 4 months, should have been mentioned earlier in the abstract and the introduction.*

**Response:** Time frames have been added to the abstract.

**Reviewer 2:** *- L195: The "ambivalence" is only a redundancy from the point of view of interpolation, but you could have exploited these crossing points as temporal information to calculate the errors related to the "quasi-stationary" assumption.*

**Response:** A good suggestion that can be implemented in subsequent publications on these data and model. It is included in the relevant section of the article.

**Reviewer 2:** *- L198: the "quasi-steady-state" assumption is only mentioned in the "Nudging" section, when you cannot avoid it any longer, although it has been implicitly a major assumption since the beginning of the paper. Please formulate it upfront in the introduction and reflect on its implications for the study.*
**Response:** "quasi-steady-state" assumption is introduced in introduction and abstract now.

**Reviewer 2:** *- L200: high drift speed compared to the water velocity. The drift speed has been set to 0.7 m/s above, the velocities of 1cm/s are only mentioned in the discussion section.*

**Response:** Thank you for pointing this out, the average drift speed during the described period was about 12 cm/s. However, a value of 0.7 was used in the parameterization of the upper boundary in the turbulent closure. Here 0.7 refers to the upper boundary condition for the dynamic wind speed in the turbulent energy budget equation. It is not the ice drift speed as it might have seemed from the article's text. This has been corrected in the new version.

**Reviewer 2:** *- L206: I can understand that submesoscale features of size 10 km located hundreds of kilometres apart are independent, but the mixed layer depths may change a lot within 4 months, please kill the suspense and indicate that this will be discussed later.*

**Response:** Line L206 is be modified.

**Reviewer 2:** *- L210: The nudging term acts on temperature and salinity but the model currents are only corrected progressively through geostrophic adjustment, which makes the model inconsistent during the adjustment time (this is - by the way - an aspect better handled by advanced data assimilation than nudging), what is the typical timescale of this adjustment in your case?*

**Response:** It takes four model months to reach a stable solution with the specified nudging coefficients. This was explained in the experiment description section and detailed in the figure with schematic of conducted simulations.

**Reviewer 2:** *- L215: Why use two different relaxation time scales for temperature and salinity? What does that mean for the dynamical adjustment of the model to density changes? Can you at least indicate the values of the two time scales (Trelax comes later, but I cannot locate Srelax in the text)*

**Response:** The relaxation time scales for temperature and salinity are the same, as mentioned in the above comment. We alter the temperature and salinity, thereby changing the density, which leads to changes in baroclinic pressure and, consequently, alterations in the dynamic characteristics of the system. Due to the gradual change in temperature and salinity, the system gradually adapts its dynamics. Trelax - Temporal relaxation coefficient (line 239). it is the same for temperature and salinity. We do not have any Srelax.

**Reviewer 2:** *- L221: Moving at 0.7 m/s, 2 minutes correspond to 80 meters (is this what you meant with "horizontal resolution"?) and are often within the same model mesh cell.*

**Response:** The actual ice drift changes over time; 80 meters is the distance between two measurements. Indeed, multiple measurements can be located within a single model cell, which is one of the reasons why we smooth high-resolution data onto a model node using the inverse distance weighting method.

**Reviewer 2:** *- L228: The mathematics of spatial interpolation have progressed significantly since 1968. Maybe the inverse distance method combined with the kd-tree does perform well, but the choice is not justified here.*

**Response:** Since our goals and objectives do not include the creation of 3D interpolated fields, and since our primary concern is the smoothness of data at the points where buoy trajectories intersect, as well as to slightly extend

the nudging area around a single node again for the smoothness of the nudging impact on the model, the specific method of distributing observations across model grid nodes is not important to us, and considering that this combination of methods is quite common in data processing for unstructured grid modeling, the use of this method seems sufficient to us, and moreover, a detailed discussion of interpolation methods is clearly out of the scope of this work.

**Reviewer 2:** - *L230: The sharp transition between the cells that do and do not participate in the nudging should be mentioned here rather than in the end of the article.*

**Response:** Grid cells in the model do not participate in nudging, only the nodes where passive tracers such as temperature and salinity are calculated. The fact that only a part of the nodes are used for nudging is mentioned on lines 217-219: "The term responsible for nudging was included only for grid nodes in the immediate vicinity of observations. To do this, a mask of nodes has been precalculated for each type of observation and is explained below." Areas with and without data are indicated in Figure 1a.

**Reviewer 2:** - *Eq (4) looks like an inverse square distance interpolation in the vertical dimension but goes to zero in the separations between observed levels at depths. This seems excessively complex in the circumstances. Not relaxing between two observed levels seems prone to inconsistencies (unstable density profiles between two observed levels), a more intuitive solution would have been to perform vertical interpolation of the SIT profiles to the model levels (linear or cubic splines), ensuring the density increases with depths, and then relax with a single coefficient. The adequacy of the vertical interpolation should be better justified.*

**Response:** We could have managed without extending the nudging area vertically and used nudging in the model only on 5 layers, but this would have required significantly reducing the computational time step to overcome sharp changes. And yes, we do not nudge to artificially interpolated data where it doesn't exist, and that is precisely why the values are zero between layers where data is present. Even if this leads to minor instabilities in the initial moments of time, the model, due to its own dynamics, mixes out this instability. Considering that such changes are local and limited to the area where nudging is performed (in a very limited number of nodes and vertical levels), it is not critical for the model's instability.

What the reviewer suggests could be tried in future work, but this would first lead to much more dangerous horizontal instabilities, when somehow interpolated data from 5 levels to a full profile would create significant baroclinic horizontal gradients leading to artificial giant velocities. Moreover, such vertical interpolation should assume the presence of background information about possible vertical characteristics of the water column, and in the case of a eddy, such smoothing simply negates any sense in further using the model, as all heterogeneities in the form of vortices will disappear because the model will be forcibly nudged to a fictional profile.

**Reviewer 2:** - *Eq(4) Is the surface temperature relaxed to the freezing point temperature or is that already handled by the FESOM model?*

**Response:** Nudging in the FESOM-C model occurs only in places where there are data, as described in the article. In our model, there is no nudging to the freezing temperature or restoration of surface salinity, as in global models.

**Reviewer 2:** *- L238: Note here that a relaxation time of one day is considered very strong relaxation in practice.*

**Response:** The relaxation time depends on the model's tasks and application area. If we are considering a global task with coarse resolution, then one day might be a large value, but for tasks with high resolution in the coastal zone, modeling tides or internal waves, the relaxation time will be significantly less. What practice is the reviewer referring to, and why is one day considered very strong? For this particular task, one day was a successful choice.

**Reviewer 2:** *- L240: I imagine that the maximum distance changes together with the maximum number of values but please specify explicitly. Also mention the size of the largest and smallest neighbourhood tested.*

**Response:** The maximum distance is defined as 1 kilometer, see line 228 of the article.

**Reviewer 2:** *- L244: The deeper profiles are nudged over shorter distances than the shallow SIT profiles, making their effect probably negligible. This is counter-intuitive since the length scales are longer at depths. Please explain.*

**Response:** SIT buoys do not provide any profiles! SIT buoys have only 5 horizons, with the maximum depth being 100 meters. See the detailed description of the instruments on lines 174 - 179 of the original article, where the maximum variability is observed. Considering that SIT buoys provide data every 2 minutes, and instruments that provide incomplete vertical profiles once a day, and complete profiles at best once a week, the effect of profiles on the nudging of surface layers is significantly less, simply due to their insignificant quantity. At the same time, zones deeper than 100 meters are determined exclusively by profiles.

**Reviewer 2:** *- L247: No reason is given why the OC/PS profiles vertical relaxation is also different from the SIT profiles. Is it because these profiles have higher vertical resolution than the model?*

**Response:** SIT buoys do not provide any profiles! For two different types of data (profiles (PS/OC CTD) and individual horizons (SIT buoys)), we have two slightly different methods of nudging, one for profiles and one for data from individual horizons.

**Reviewer 2:** *- L247: The model is nudged towards invariant temperature and salinity fields interpolated from the SIT profiles. These interpolated maps being the only external forcing of the model, they should be shown at a representative depth, for example the salinity above the halocline (20m or 50m) and the temperature at 100 m.*

**Response:** Refer to the above response about interpolating data across the entire space, and that nudging occurs only within a distance of up to a kilometer from the data. In spaces between buoys, there is a lack of data and interpolation, so there are no two-dimensional maps as such. The only thing that can be shown is scattered plots of data, examples of which are shown in the data links.

**Reviewer 2:** *- L249-250: These technical details can be removed. - L256: The "small time resolution" of the instruments should probably be called "low frequency sampling".*

**Response:** changed in revised manuscript.

**Reviewer 2:** *- L260: The PS-CTD profile in Figure 1 is not a typical profile as it is both warmer and more saline than all the other profiles, so it is not obvious if this profile is overall more stratified or more unstable than the profiles that will be assimilated later. If you plot all density profiles as thin lines, you can give an indication if the assimilation will have a stabilising or destabilising effect overall.*

**Response:** Thin profiles refer to profiles from the MSS instrument. Since most of MSS measurements are concentrated in the northern part of the drift, they stand out to one side of the profile that was used for the initial conditions. Figure 1 was changed.

**Reviewer 2:** *- L265: It is very well that the authors admit the breach of the continuity principle, but this should have been admitted earlier in the methods description. More advanced multivariate data assimilation methods may mitigate that problem, which could be worth noting in the discussions. Alternatively, the authors could have calculated geostrophic current velocity increments from the density gradients caused by the nudging term, by analogy with the Cooper and Haines (1996) method.*

**Response:** We are unable to calculate geostrophic current velocity using the nudging term, as continuous density fields from observations are not available to us. Additionally, we lack interpolated fields derived from the data (as mentioned earlier). However, it's important to note that our use of Data Nudging is restricted to specific observation locations, rather than being applied across the entire area. This localized application prevents any significant issues during the phase of determining initial conditions for a free simulation. The corresponding changes have been made to the manuscript.

**Reviewer 2:** *- Figure 4 sections show discontinuities in the vertical salinity profiles, are these real or are there only a few bands in the (very small) colour scale?*

**Response:** It's unclear which discontinuities are being referred to. If the discussion pertains to Figure 4a, then the data are linearly interpolated between profiles from observations, as noted in the figure description.

**Reviewer 2:** *- L286: This sentence is a very contorted way to acknowledge the temporal evolution of the ocean variables. Again, it is regrettable that the cross-over differences have not been exploited as mentioned earlier.*

**Response:** The sentence is rewritten in the revised manuscript. In such cases, the model points are aligned with at least two separate observations of the same variable at the same location, highlighting the limitations of the quasi-stationary approximation assumption. Typically, the model strives to replicate the smoothed values derived from these overlapping observations.

**Reviewer 2:** *- L297: "Same order variability at the model grid scale" this variability is not visible in Figure 6, has it been smoothed?*

**Response:** Thank you, that was a poor formulation. Removed.

**Reviewer 2:** *- Section 3.2 T/S reconstruction contains lengthy descriptions. Please reconsider if they can be shortened. - L300-301: Unclear sentence about the slope being "deeper than 40 m". Please rephrase.*

**Response:** Rephrased: Both sections reveal an increase in the Mixed Layer (ML) depth from northwest to southeast. However, the slope of the isopycnals from west to east is less consistent below approximately 40 meters compared to the north-south section.

**Reviewer 2:** *- L304: Why is the bulge associated with mesoscale features? - L306: "characterise the simulated system as isotropic". Unclear as well, please rephrase.* **Response:** Removed, this analysis is beyond the scope of the article.

**Reviewer 2:** *- L315: Only here do the authors first admit that the EKE collapses by construction of the model. Knowing that the nudged EKE is resulting from an interpolated dataset containing both temporal and spatial variations and not strictly "steady state", I would expect that the resulting EKE would initially be too high. Remember that the interpolated fields are not shown so the readers are free to imagine what has come in there. So please show the interpolated fields and indicate - even roughly - the a priori expected range of values of EKE that should be reasonable.*

**Response:** The absence of interpolation fields has already been noted earlier. The model is not nudged to EKE, only to temperature and salinity. From the observed profiles of temperature and salinity, we cannot estimate EKE. For comparison with the expected values of EKE, references to relevant works are provided.

**Reviewer 2:** *- Figure 7a) shows tiny mesoscale features but a lot of the areas are white. It should be rotated (the X and Y axes have not meaning anyway) and cropped to maximise the useful area.*

**Response:** The X and Y axis labels are switched (an error in the figure). The figure was cropped to fit all elements, including the area over which averaging was done in stereographic coordinates. In geographic coordinates, it's difficult to assess scales.

**Reviewer 2:** *- Figure 7b) I cannot see the yellow solid lines. Try making them thicker.*

**Response:** The figure has been redone.

**Reviewer 2:** *- Figure 8 also has too much white area. Rotate and crop for clarity.*

**Response:** The figure has been redone, but not rotated. These are stereographic coordinates; rotating it would only confuse its orientation relative to Figure 1.

**Reviewer 2:** *- Figure 9 is very nice and even shows internal waves that are not discussed in the text. This could be added if space permits.*

**Response:** The figure pertains to the dynamics of individual eddies; a discussion of internal waves would require a separate section.

**Reviewer 2:** *- L345-349: Is the description of the eddies movements necessary for the rest of the paper?*

**Response:** In our opinion, yes.

**Reviewer 2:** - *L359-361: This argument does reach any conclusion, so I will give you mine. The ice drift change direction in the Northern part but the model forcing was constant, however the data coverage is more complete where the ice drift is sinuous. This means that the data sampling affects the simulated vortices, which should be more abundant to the top part of the graphs.*

**Response:** Thank you for your contribution, it is included in the new version of the manuscript: "In the northern part, the ice drift changes direction, but the model forcing remained constant. However, data coverage is more comprehensive where the ice drift is sinuous. This implies that data sampling influences the simulation of vortices, which are expected to be more prevalent in the upper portion of the graphs."

**Reviewer 2:** - *Section 4.3 "Method limitation" does not discuss much the limitations arising from the results but mostly limitations by construction that could have been flagged upfront in the method description instead of leaving the readers wonder about them throughout the paper. Paragraph 369-378 is probably the only proper discussion of the results and should stay there.*

**Response:** The section is divided into two parts; the first part has been moved to the methods section.

**Reviewer 2:** - *L374: What do you mean by "displaced"?*

**Response:** mixed layer. changed.

**Reviewer 2:** - *L379-385: According to the previous paragraph, the large-scale gradients can be trusted as "instantaneous" (or synoptic), but not so much the small scales. So a solution following Robinson et al. 1996 could be beneficial here: the large-scale component of the interpolated data can be used for nudging, while discard the small-scales, which can be excited by random mesoscale perturbations all over the model domain, thereby removing the "internal boundary".*

**Response:** We executed the process in two steps. First, we reconstructed with a coarse resolution (1 km mesh), then with a high resolution (250 m), as described in the experiment section, avoiding interpolation. In contrast to random (unknown) mesoscale perturbations, we utilized actual data.

**Reviewer 2:** - *Same paragraph: The same remark applies to the vertical interpolation since you have used a similar square distance function with only 3 meter characteristic depths. An "internal boundary" in the vertical may a priori have more adverse effects on this study.*

**Response:** As noted in the manuscript and previous answers, we do not use interpolation between layers. Therefore, data from 75 m depth have no influence on the nudging at 50 m depth.

**Reviewer 2:** - *L384-395: I thought that you already did a sensitivity analysis to the influence radius, did you not test a larger radius?*

**Response:** You are correct. It was a leftover from the draft version. The sentence has been removed to avoid confusion.

**Reviewer 2:** - *L388: It is clear that 1cm/s is much smaller than the 0.7 m/s ice drift but that could be noted upfront. The general direction of the currents could be indicated as well for information.*

**Response:** An example of the velocity distribution is shown in Figure 8. It is extremely difficult to indicate the main direction of the currents in this case.

**Reviewer 2:** *- L410 The number 630.000 may seem quite impressive but still does not make a proper synoptic measurement campaign. Please acknowledge that.*

**Response:** To avoid a discussion about oceanographic synoptic measurements which is unnecessary in this manuscript, we prefer to omit the comment on non-synoptic measurements in this context, especially since it would contradict the overview of this expedition: Overview of the MOSAiC expedition: Physical oceanography. Elem Sci Anth, 10: 1. DOI:https://doi.org/10.1525/elementa.2021.00062.

**Reviewer 2:** *- L412 As noted earlier, the measurements are not independent if they are located within one Rossby radius of the assimilated profiles. There are spatial autocorrelations that reduce the significance of the validation.*
**Response:** The response regarding the independence of data has already been given earlier.

**Reviewer 2:** *- L417: I would not claim that these are "dynamically consistent" as long as the measurements are collected along a 4-months trajectory across moving eddies. You have criticised the "quasi-stationary" assumption earlier so you should moderate this claim accordingly.*

**Response:** Corrected.

**Reviewer 2:** *- L421-422: Here would be the adequate place to recap these insights. I can note the vertical maxima of EKE in the Atlantic layer and the halocline. These findings may have been noted by earlier studies but it is still good to confirm or contradict earlier papers.* **Response:** Summary is extended.

**Reviewer 2:** *- Code and data availability: Are the in situ profiles publicly available?*

**Response:** All MOSAiC expedition data are publicly available and open. The data can be reached by the links in section 2.4 Observational Data, or visit the PANGAEA database and also view data on the ITP profiles on the WHOI website.

**Response: All noted typos are addressed in the revised version of the manuscript.**

*Typos:*

- l28: add a comma between Basin and Zhao.

- L. 80: DN is undefined at this point.

- l82: close the parenthesis after Fang et al. 2023 (submitted).

- L.121: Li et al. misses a year.

- L138: Define ML as Mixed Layer.

- Eq (1) indicate that z is the depth, positive downwards.

- Figure 4: The subplots labels a, b, and c are wrong in the caption. They should be b, c, and d.

- L303 "low-salinity (high-density) intrusions": should this rather be "low-density"?

- L340 "and about 5 km" is missing an "is".

- L425: "The rest".

- The reference to Sokolovskiy and Vernon has duplicate title but no journal name nor volume number.

References :

Bourgain, P., & Gascard, J. (2011). The Arctic Ocean halocline and its inter-annual variability from 1997 to 2008. Deep Sea Research Part I: Oceanographic Research Papers, 58(7), 745–756. https://doi.org/10.1016/j.dsr.2011.05.001

Cooper, M., & Haines, K. (1996). Altimetric assimilation with water property conservation. J. Geophys. Res, 101, 1059–1077.

Robinson, A. R., H. G. Arango, A. J. Miller, A. Warn-Varnas, P.-M. Poulain, and W. G. Leslie (1996), Real-time operational forecasting on ship-board of the Iceland-Faeroe Frontal variability, Bull. Am. Meterol. Soc.,77, 243–259.

Troupin, C, Barth, A, Sirjacobs, D, Ouberdous, M, Brankart, J.-M, Brasseur, P, Rixen, M, Alvera Azcarate, A, Belounis, M, Capet, A, Lenartz, F, Toussaint, M.-E, & Beckers, J.-M. (2012). Generation of analysis and consistent error fields using the Data Interpolating Variational Analysis (Diva). Ocean Modelling, 52-53, pp. 90-101.

Zimmerman, D., Pavlik, C., Ruggles, A. et al. An Experimental Comparison of Ordinary and Universal Kriging and Inverse Distance Weighting. Mathematical Geology 31, 375–390 (1999). https://doi.org/10.1023/A:1007586507433

---

## Referee Report (RR1)

*Review of the manuscript entitled « Dynamical reconstruction of the upper-ocean state in the Central Arctic during the winter period of the MOSAiC expedition" by Kuznetsov et al..*

The authors present a reconstruction of the dynamics based on the nudging of the MOSAiC data in a high-resolution model (FESOM-C). The goal of the paper is to demonstrate the usefulness of this modeling tool to analyze the MOSAiC data and to give a better description of the mesoscale and sub-mesoscale dynamics in the central Arctic.

The authors have developed an interesting tool to interpret the MOSAiC dataset, however, the analysis of the model simulation is limited and could be more detailed. The authors point to the bimodal vertical distribution of the EKE. As mentioned by the authors this bimodal distribution was already described in previous studies, the authors should specify the novelty of their result. For example, the origin of the north-south distribution of the EKE could be detailed. The authors describe the properties (size, depth, …) of an anticyclonic eddy and a cyclonic eddy and their interaction. The authors might extend this analysis to all the eddies of the area to give a broad view of the distribution of the properties of the eddies and discuss how it compares with previous studies. Furthermore, the interaction of the eddies remains quite qualitative and might be more detailed (implications for the evolution of the properties, …).

**Specific comments.**

L 162. The initialization of the coarse resolution model could be specified in this paragraph.

L. 219-220: I do not really understand this sentence. Could the authors clarify?

L. 248: distance along the vertical?

L. 251-252. What do the authors mean by similar manner? According to the authors response to reviewers, I thought that the $C_2$ was constant with depth. Is it correct?

L.253: The model is nudged to ITP profiles in the same way as the PS and OC-CTD profiles?

L 256. Add a reference to fig 2e.

Fig 2. 2b: $d_i$ is the inverse of the distance?

L. 289: The duration of the free run is mentioned in Fig 3, but it should also be specified in this paragraph.

Figure 5. fig 5b and 5c have been inverted? Check the legend: Blue line is salinity and orange line temperature.

Section 3.2. This section has to be checked carefully. North and East have been inverted.
L.314-315: "a decrease in ML salinity …. spatial variability"? This sentence is unclear to me, could the authors clarify?
L.317: Could the authors indicate the halocline depth.
L.318: "ML depth increases". I would rather say that the ML decreases. Is it correct?
Figure 6: Check the legend (East and north inverted).
L. 319 "Increase". Change in decrease?
L. 323. "Low salinity (high density)"?

L. 343-344. Could the authors discuss the origin of the difference of the EKE distribution between the northern and southern parts of the domain?

Figure 7: The figures are not easy to read. Larger plots might help.

Figure 8 What do the figure 5, 10, 15, 20 mean? Could the authors label the anticyclonic and cyclonic eddies that are discussed?

L. 370: "obsevations" : observations.

L. 371. Could the authors correct the sentence?

---

## Author Response (AR2)

**Response to Reviewer 1 for the Paper (Second Iteration):** "Dynamical reconstruction of the upper-ocean state in the Central Arctic during the winter period of the MOSAiC Expedition."

**Reviewer 1:** *The authors have greatly improved the manuscript and only minor further revisions are necessary. The revisions improve the readability of the manuscript and by and large have addressed the concerns of my previous review. A few points remain that should be addressed before the manuscript is suitable for publication.*

*While the introduction is much better than it was before, the tendency to list previous results paper-by-paper rather than synthesizing the important findings remains, particularly from lines 26 to 53. The style later in the introduction, such as from lines 64 to 97, is much better, with relatively few instances of listing previous results and more examples of synthesis.*

**Response:**
We have revised this part of the introduction, condensed and reorganised the text. Please see the version of the manuscript with all changes marked for the detailed changes.

*Regarding the MSS data, some qualification is needed for the use of the term "independent". I understand that the MSS data was not included in the nudging process, and so in that sense it can be considered independent. However, geospatial measurements typically have very strong autocorrelation structures, so measurements within the decorrelation length scale cannot be considered statistically independent. In my opinion, a caveat discussing the nature of independence is needed. The authors could address this issue with an indication of the spatial structure of correlation within the observational network. At minimum acknowledgement that the data is likely autocorrelated is needed.*

**Response:**
Following part was added to the manuscript:
"The MSS data, while not included in the nudging process and thus considered independent to a degree, inevitably exhibits some degree of autocorrelation with DN and PS measurements. This is particularly due to the spatial distances between DN buoys and the temporal and spatial dispersion of data from PS and MSS. Consequently, we acknowledge the data as independent with the caveat that a certain level of autocorrelation is indeed present, reflecting the inherent spatial and temporal structures within the observational network."

*Minor changes refer to lines in Version 3 of the manuscript (i.e., not the version with tracked changes.)*

*Line 5 – should FESOM-C be in parenthesis? (Otherwise the abstract looks good to me!)*

**Response:**
Corrected as recommended.

*Line 21 – minor change needed in the list order, since the grammar is am-*

*biguous. As written it could be interpreted as saying that "heat" is part of the list of "organic and inorganic matter", which I'm sure is not what was intended.*
**Response:**
Corrected.

> *Line 30 – Parenthesis around reference.*

**Response:**
Corrected as recommended.

> *Line 32 – Capitalize "v" in von when it's at the beginning of a sentence.*

**Response:**
Corrected.

> *Line 53 – missing end of sentence.*

**Response:**
Corrected.

> *Line 149 – "Mixed Layer" does not need to be capitalized.*

**Response:**
Corrected.

> *Line 174: "Most" does not need to be capitalized.*

**Response:**
Corrected.

> *Line 185: "was used" → "were used"*

**Response:**
Corrected.

> *Line 207-8: Sentence is unclear. Perhaps it should say "observational data nudged the model at the same" rather than "observational data nudged by the model"?*

**Response:**
Corrected.

> *Line 286 – "lead" → "can lead" or "lead" → "leads" (probably the later).*

**Response:**
Corrected.

> *Line 310 (and throughout) – Root mean square error is typically abbreviated as RMSE not RMSe. Unless there is a strong motivating reason for this capitalization scheme, the standard all-caps version should be used.*

**Response:**
Corrected.

> *Line 323 – grammar is unclear.*

**Response:**

Corrected.

*Line 409 – reanalysis data using MOSAiC data are already available (ERA5), however I don't know whether any ocean reanalysis data is available yet.*
**Response:**
To the our knowledge, the MOSAiC data were used only in atmospheric models. According to ECMWF support (request dated 18 March), no MOSAiC data were used in the ocean reanalysis.

*Line 427 – "profiles were nudged by the model" → "profiles were used to nudge the model."*
**Response:**
Corrected.

**Response to Reviewer 3 for Paper:** "Dynamical reconstruction of the upper-ocean state in the Central Arctic during the winter period of the MOSAiC Expedition."

*Review of "Dynamical reconstruction of the upper-ocean state in the Central Arctic during the winter period of the MOSAiC Expedition" by Kuznetsov et al.*

*The study aims to use the FESOM-C model to create a high-resolution gridded set of ocean fields in the Central Arctic based on MOSAiC data and other complementary data. A nudging method is developed that uses the observational data, and the model is then validated against independent data. The resulting outputs are used to infer EKE and individual eddy behaviour in the region.*

*I believe that a lot of work has gone into the study, and it will certainly provide a strong base for future studies of the region. Therefore I do not think that further analysis is required. However, I found that the presentation of the study needs to be much clearer. The study attempts to be both a methods paper and an analysis study, which means that it must put significant effort into emphasising what is a new method, what is validation, and what is a result. This was only apparent to me after reading the paper multiple times. I think one key thing to explain better is when the nudged or free-run outputs are being used, and if it is the free run, what day it is, and how that day represents the state away from the nudging.*

*I don't doubt that efforts have gone into trying to make the manuscript clearer (something brought up in the last round of reviews), but believe that more work needs to be done to highlight to the reader what each section is doing and what the outcomes from each are. I also found a number of incorrect captions/figure labels (detailed at the end of the review), which made the text harder to understand. Below I have listed issues with each section that, if fixed, should help with the clarity. This is then followed by line-specific comments.*

**Response:**
We are grateful for your detailed analysis of our article and for your substantial and pertinent observations. We have taken all your comments into account and have implemented the necessary modifications to the manuscript. Below are the responses to each issue you've raised. We believe these revisions will significantly enhance the clarity of our work and appreciate your invaluable input in making these improvements.

*Abstract: The description of the results (lines 11-13) is confusing (and I have also noted this in the results later). Stating "in that direction" and "opposite characteristics" is too vague when there is no figure or discussion for context. It is better to be explicit, for example, "we do find an increase in mixed layer depth from west to east" and "whereas in the south-north direction, it deepens". I also*

*think that using "increase" to describe mixed layer depth is very confusing; it implies a deepening but seems to be used here as a shallowing. I would suggest to use "deepening" and "shallowing/shoaling" to refer to what happens to the mixed layer depth, to avoid ambiguity.*

**Response:**
Abstract corrected according to reviewer's suggestions

*Introduction: I found the introduction very long. There is a lot of text about eddies, which is only one part of the paper. I understand that you want to emphasise that a high resolution simulation is required in order to understand them, but five paragraphs is a lot and shifts the weight of the paper away from the part of the introduction that describes the need for the new method. I think making the introduction more concise would greatly help the reader to understand what you are doing and why.*

**Response:** We have revised this part of the introduction, condensed and re-organised the text. Please see the version of the manuscript with all changes marked for the detailed changes.

*Methods: - In section 2.3, it would be useful to refer to figure 3 to emphasise the use of each model setup in each stage of the process.*

**Response:**
We appreciate reviewer's suggestion and have updated the manuscript to include references to the detailed description of the experiments in section 2.6 and Figure 3, as was recommended.

*- I think it would also be more useful to put the observational data as the first subsection, so that all the model sections remain together – this would help the flow.*

**Response:**
The subsection on observational data was moved to the beginning.

*- Section 2.5: The nudging is one of the main points of the work, so a sentence emphasizing that at the start of the subsection would be useful. I found the first paragraph was overly complex and could do with some rephrasing. The assumption of quasi-steady-state is a big one and it needs to be stated upfront that this will have caveats, rather than much later on. I am also a bit confused about the example of the depth of the mixed layer being affected by the fact the data is over 4 months... the mixed layer is a quantity based on the temperature and salinity, so if that is varying, surely that suggests the temperature and salinity will vary non-negligibly too?*

**Response:**
The paragraph was partially rewritten and rephrased, taking into account the comments.

The ML is undoubtedly defined by salinity and temperature; changes over time in the salinity to which we are nudging lead to changes in the ML. However,

the expected effect works in the opposite direction, meaning the gradient should presumably be stronger than we reconstructed. We note this at the end of the article after analyzing the results.

*- The last paragraph of 2.5 was confusing. Are you saying that they should be included but aren't directly, or they shouldn't be included but are? Would your assumption of the ice drift velocity (and associated stress) fail in this situation? What are the implications of that? How will the inclusion of storms affect nudging data that also takes data points that did not experience a storm?*
**Response:**
The paragraph was rewritten and rephrased.

*- Section 2.6: I think figure 3 should be explained more thoroughly here. I had to refer back to it to understand when the free run was being used and why. Could you make it clearer what will be used for validation and what will be used for analysis? In the following sections, you use outputs at various times in the free run, and sometimes it is not stated what time is being used. I see the point in comparing the start and end of the free run (as in Figure 4) to check its evolution, but each time you use a given output in subsequent analysis you should explain why that one was chosen.*
**Response:**
The section was expanded to include a more detailed explanation of the experimental scheme and to provide clarification on which simulations were utilized for analysis. "To reduce computing time, the initial run with nudging was conducted on the coarse mesh (1 kilometer) ... .
The duration of the free run was 19 real days. Results of the free run ...
In the following, we used the results from high-resolution mesh ... "

Results: *- Section 3: I think there should also be a demonstration of the temperature fields in Figure 4, since your main outputs from the model are both salinity and temperature*
**Response:**
In accordance with the reviewer's comment, a figure comparing the temperature from independent data with model results has been added, and the corresponding references have been included in the text.

*- What model output does Figure 6 show? Is it the free run after 2.5 days, as in Figure 5? This should be stated in the text and the caption*
**Response:**
The corresponding text and caption have been added.

*- Please be very careful about how you use the word "increase" in relation to the mixed layer depth. In the abstract and this section, it seems that "increase in ML depth" is used to mean "shallowing". This is very misleading. It is better to use "deepen" and "shallow/shoal"*
**Response:**

Corrections have been made.

*- There was not a demonstration of the reconstruction in between the nudged data locations. I think it would be useful for the reader to see some 2D maps or similar to show what the fields look like spatially (and how much the nudging is affecting the surroundings). For example, since the ML depth is described as varying north-south and east-west, it would be useful to see how this looks on a map in the regions away from the nudging locations, and at different times of the free run. It would then be clear how much the nudging has forced the model from the initial conditions.*
**Response:**
It's difficult to agree with the reviewer's comment in this instance. Reconstructions within the data domain and between observation positions are demonstrated in Figure 8 - the horizontal salinity field, and in vertical sections in Figure 6. Given that the initial conditions are represented by a single profile across the entire area, changes induced by nudging are deviations from the constant in the horizontal section. We do not display data outside the nudging zone; what would be the purpose? We can only reconstruct within the data domain and in the immediate vicinity of the data, as illustrated in the figures.

*- At the end of this section, since this is one of the main aims of the paper, it would also be nice to have a summary statement of how your model is behaving to ensure it is reasonable enough to proceed with.*
**Response:**
The corresponding statement has been added to the end of the Model Validation section. "In conclusion, following the model validation, our comparison with independent data indicates that our method yields sufficiently accurate results. Therefore, it can be reliably used for the reconstruction of three-dimensional fields."

Discussion: *I was under the impression that the point of the nudging and then free run was to get a (quasi-)steady-state reconstruction of the ocean, so it was a little strange to me that the free run was being used to analyse evolution of eddies in the absence of external forcing in section 4.2. I think the motivation for doing this is important to state at the beginning of the subsection. I acknowledge that its use is somewhat explained from line 369 onwards, but an introductory statement and justification would be useful*
**Response:**
Clarifications and motivation have been added at the beginning of the section "Eddy examples". "As has already been noted, the system achieved a stable numerical solution by the end of the period when the model is nudged towards the data. However, after the external force in the form of nudging is removed, the system begins to change. By examining the changes during the free-run, one can study the dynamics of the formed eddies."

Summary: *For a paper that develops a method, I found it strange that there*

*was no reflection on how the method could be used further in the future (either developing this work, or for others to use). I think having such a reflection would help emphasise the main uses of the study.*

**Response:**

Two paragraphs were added at the end of the summary. "This study presents ... . For future development, ... . "

*Lines 125-127: what is the minimum depth of the datasets? I know that the SIT buoys are 10 metres, but what about the others? Does this adversely affect the nudged ML salinity and temperature?*

**Response:**

The minimum depth of the ITP profilers varied from 5 to 8 meters, while the minimum measurement depth for the PS/OC CTD was 2 meters. Considering that the ML depth was over 20 meters, it can be asserted that the temperature and salinity of the ML were well represented in the data.

Text was added to "FESOM-C model" section: "The minimum depth of the observational data from instruments ranged from 2 to 10 meters. Considering that the mixed layer depth exceeded 20 meters, it can be asserted that the temperature and salinity within the mixed layer were well represented in the data."

The descriptions of the instruments were also modified.

*Line 140: how does this assumption hold across the full domain? Later on, it is acknowledged that there were different ice conditions in different regions – does that affect the results? A sentence or two about the limitations of this assumption would be useful. For future applications, is it possible to use this method with different ice conditions spatially?*

**Response:**

Since we employ a stationary approximation, the boundary remains constant. Undoubtedly, leads and storms play a role in altering mixing, but as we noted in the article, their impact was accounted for through nudging to observational data.

Utilizing this method under different ice conditions spatially is also feasible. In the case of a non-stationary approach, such as when applying atmospheric forcing, the upper boundary condition determined by ice and atmospheric dynamics would accordingly change. Similarly, in other modeling tasks, for instance, modeling leads, appropriate boundary conditions for turbulent closure can be set as constants or varied spatially. This is a relatively standard approach for defining the upper boundary in turbulent closure.

The varying ice conditions in this task (winter period, central Arctic) should not play a decisive role in mixing; the ice drift speed plays a significantly more crucial role here.

A sentence added to the article reads: "Since we use a quasi-steady-state

approximation (see the nudging section **??**), this parameter remains unchanged throughout the entire computation process, although it does not describe individual storm or lead events. We compensate for these with model nudging to observations."

*Line 195: please state upfront that the movement is southeast to northwest, rather than the reader needing to infer it from the buoy trajectories*
**Response:**
"During this period, the MOSAiC expedition drift direction was from the southeast to the northwest." added to "Observational data" section.

*Line 202: "ambivalence" is a strange word to use. Maybe use "uncertainty" or similar*
**Response:**
corrected to "uncertainty"

*Line 207: this should be rephrased to "all observational data used to nudge the model"*
**Response:**
The whole paragraph was rewritten according to previous suggestions.

*Paragraph starting line 253: how much is the model affected by having lots of data for nudging above 100 metres and less data to nudge below that?*
**Response:**
The text was added to the end of the following paragraph: "The dynamics activity and variability in the upper 100 meters of the ocean are significantly higher compared to deeper regions. The abundance of data in this upper layer allows for a detailed representation of submesoscale processes, leveraging the system's dynamic nature. Conversely, the deeper zones exhibit less variability, making them amenable to accurate representation with fewer data points. This differential data density aligns with the varying dynamical characteristics of these oceanic layers, ensuring the model's efficacy across depths."

*Lines 268-270: this is a repeat of lines 225-227. While I appreciate the reiteration of this caveat, I do not think it needs to appear twice in the same subsection*
**Response:**
The whole paragraph was rewritten according to previous suggestions.

*Line 281: I think this should be section 2.3*
**Response:**
Changed.

*Line 286: "leads" or "led"*
**Response:**
Corrected.

*Line 302: "dissolves eddies" is a strange term*

**Response:**

Changed to "dissipates"

*Line 299: state that here you mean "free run after 2.5 days" when you say "model"*

**Response:**

"model" changed to "free run after 2.5 days"

*Line 311: what depth is that in this region?*

**Response:**

depth is about 4500 meters. following sentence was added to the "model domain" section: "The model domain covers the entire water column, reaching a maximum depth of 4450 meters, which represents the average depth for this region."

*Line 314: I think you mean 115 E, 86.2 N*

**Response:**

yes. corrected.

*Line 319-320: it is not necessary to say "both sections reveal. . . " here, as you have just described one of them in the previous sentence. Suggest rewording this and the previous sentence to avoid repetition*

**Response:**

sentence removed.

*Line 323: why is "high-density" in brackets? The way it is written implies that it is synonymous with "low-salinity". I would suggest rephrasing*

**Response:**

rephrased: "In reality, low-salinity intrusions into the ML from the surface can be attributed to changes in both surface heat and salt fluxes. However, in this study, the influence of these fluxes is simulated by nudging, suggesting that the submesoscale variability of the ML depth is most likely governed by eddy dynamics."

*Line 340: I was under the impression that your assumptions of the surface friction would also prevent you from studying an ocean that is experiencing a changing ice cover. It would be good to state this (or, if I am wrong, state the converse) – as others may wish to use this method in the future and need to know the caveats*

**Response:**

In the "FESOM-C model" section we have: "... The effect of sea ice presence on the dynamics of the ocean surface layer has been parameterized by the friction between ice and ocean. Thus, we do not take into account the additional transfer of momentum due to ice drift. The effect of ice drift has been accounted for in the turbulence closure... ". To adapt to different ice conditions with the

quasi-steady state assumption, the same method can be applied. For situations that are not steady-state, full ice model coupling should be used.

*Line 371: this needs rewording*
**Response:**
"This is due to the quasi-steady nature of the eddy during the time the when drift passed this geographic position." change to "This is attributed to the quasi-steady nature of the eddy at the time when DN passed through the eddy position."

*Lines 379-385: the point about the ice drift changing in the northern part is repeated. The paragraph is quite confusing to read - I do not get how the "model forcing remained constant" fits into the argument, for example. Would suggest rewording or reordering*
**Response:**
We rephrased this paragraph and removed the references to data coverage, which is considered in Section "4.3 Limitations of our method".

*Line 427: "were nudged by the model" -¿ "were used to nudge the model"*
**Response:**
Corrected.

*Figure 1: - Please make figure 1a much bigger! It is very hard to see all of the information and boxes; the magenta box is barely visible even when zooming in. - Figure 1c is nice for context but it is still hard to know exactly where in the Arctic it is - a subplot showing the location on a more zoomed-out map with some sort of land mass would be more useful than just the bathymetry.*
**Response:**
Figure 1 was split into what are now Figures 1 (previously parts a and c) and 4 (previously part b), resulting in an enlargement of the figure. Additionally, Figure 1c has been modified.

*- Adding a direction to the drift would greatly help the reader in the following text*
**Response:**
"... During this period, the MOSAiC expedition drift direction was from the southeast to the northwest." added to the "Observational data" section

*Figure 2: where is the caption for d)?*
**Response:**
Corrected.

*Figure 4: I think the labels are wrong – a) is repeated twice, and there is no d). Why was temperature not shown? It would be very useful to see the temperature evolution in the free run, since its vertical distribution is different from that of salinity*
**Response:**

Labels corrected. Figure with temperature added.

*Figure 5: I believe that b) and c) are the wrong way round. Also, for the standard deviation plot, the caption states it is the model with nudging and the free run after 2.5 days, while the figure legend says model and obs. Which comparison is it? And for the RMSE, when you say model, do you mean nudged or free run?*
**Response:**
Corrected.

*Figure 7: Please make this much bigger. I had to zoom in on a PDF and it was still hard to see the details of a)*
**Response:**
Corrected.

*Figure 9: I think the "cyan" box is now magenta?*
**Response:**
Corrected.

*Table 1: is the free run from day 2.5? it should be stated in the caption*
**Response:**
Corrected.

**Response to Reviewer 4 for Paper:** "Dynamical reconstruction of the upper-ocean state in the Central Arctic during the winter period of the MOSAiC Expedition."

**Reviewer 4:** *The authors present a reconstruction of the dynamics based on the nudging of the MOSAiC data in a high-resolution model (FESOM-C). The goal of the paper is to demonstrate the usefulness of this modeling tool to analyze the MOSAiC data and to give a better description of the mesoscale and sub-mesoscale dynamics in the central Arctic.*
*The authors have developed an interesting tool to interpret the MOSAiC dataset, however, the analysis of the model simulation is limited and could be more detailed. The authors point to the bimodal vertical distribution of the EKE. As mentioned by the authors this bimodal distribution was already described in previous studies, the authors should specify the novelty of their result. For example, the origin of the north-south distribution of the EKE could be detailed. The authors describe the properties (size, depth, . . . ) of an anticyclonic eddy and a cyclonic eddy and their interaction. The authors might extend this analysis to all the eddies of the area to give a broad view of the distribution of the properties of the eddies and discuss how it compares with previous studies. Furthermore, the interaction of the eddies remains quite qualitative and might be more detailed (implications for the evolution of the properties, . . . ).*

**Response:** We deeply appreciate the thoughtful and detailed review of our manuscript. Your comments and suggestions have provided a valuable perspective on our work and its presentation.

We acknowledge your recommendation to expand our analysis and explore additional avenues. Indeed, your suggestions to deepen the examination of the bimodal vertical distribution of EKE and to extend the analysis to encompass all eddies in the area are both intriguing and valuable. We also recognize the importance of providing a more detailed discussion on the interactions of eddies and their implications.

However, after careful consideration, we believe that the current scope of our manuscript provides a substantial and coherent analysis that aligns with our initial objectives. We fully agree that exploring the north-south distribution of EKE and a comprehensive analysis of all eddies' properties would enrich the understanding of the Arctic's dynamical processes. Nonetheless, such an expansion would significantly broaden the scope of our current study and could potentially dilute the focus on the demonstrated usefulness of our modeling tool.Additionally, we would like to clarify more precisely that the main idea of the paper is methodological, and the eddy structures are just one of the examples that can be applied to it.

In response to your specific comments, corrections and adjustments have been made as suggested. These changes have strengthened the manuscript and clarified the points of concern.

Regarding the broader expansions you recommended, we believe these indeed represent valuable directions for future research. We are currently considering

these for potential separate studies that would build on the foundation laid by the current work. This approach will allow us to maintain a clear focus in the present paper while dedicating the necessary time and resources to thoroughly explore these complex and interesting aspects in subsequent publications. Thank you again for your constructive feedback.

**Specific comments.**

*L 162. The initialization of the coarse resolution model could be specified in this paragraph.*
**Response:** A sentence about the initial conditions for the coarse grid was added to the text.

*L. 219-220: I do not really understand this sentence. Could the authors clarify?*
**Response:** The sentence was simplified. We added an explanation of Einstein summation rule.

*L. 248: distance along the vertical?*
**Response:** yes. "along the vertical" is added.

*L. 251-252. What do the authors mean by similar manner? According to the authors response to reviewers, I thought that the C2 was constant with depth. Is it correct?*
**Response:** The sentence "The observational ... similar manner." was deleted to avoid confusing the reader. "ITP profile" was added in the next sentence. Yes, C2 is a constant.

*L.253: The model is nudged to ITP profiles in the same way as the PS and OC-CTD profiles?*
**Response:** Yes. "ITP ..." was added.

*L 256. Add a reference to fig 2e.*
**Response:** added

*Fig 2. 2b: di is the inverse of the distance?*
**Response:** di is a distance.

*L. 289: The duration of the free run is mentioned in Fig 3, but it should also be specified in this paragraph.*
**Response:** "The duration of the free run was 19 real days." added to the text.

*Figure 5. fig 5b and 5c have been inverted? Check the legend: Blue line is salinity and orange line temperature.*
**Response:** inverted back

*Section 3.2. This section has to be checked carefully. North and East have been inverted.*
**Response:** N was changed to E and vice versa.

*L.314-315: "a decrease in ML salinity .... spatial variability"? This sentence is unclear to me, could the authors clarify?*
**Response:** The sentence was rephrased.

*L.317: Could the authors indicate the halocline depth.*
**Response:** Here, the halocline depth coincides with the ML depth. For clarity, halocline has been changed to ML.

*L.318: "ML depth increases". I would rather say that the ML decreases. Is it correct?*
**Response:** "decreases" - corrected.

*Figure 6: Check the legend (East and north inverted).*
**Response:** N was changed to E and vice versa.

*L. 319 "Increase". Change in decrease?*
**Response:** "decrease" - corrected

*L. 323. "Low salinity (high density)"?*
**Response:** The sentence was rephrased.

*L. 343-344. Could the authors discuss the origin of the difference of the EKE distribution between the northern and southern parts of the domain?*
**Response:** Unfortunately, with our method, it does not seem possible to reconstruct the genesis of these eddies. Consequently, any discussion of the possible reasons for the differences would either be extremely superficial or speculative, which we would like to avoid in this article. Undoubtedly, this is an important question that can be addressed in future work.

*Figure 7: The figures are not easy to read. Larger plots might help.*
**Response:** Images rearranged for enlargement.

*Figure 8 What do the figure 5, 10, 15, 20 mean? Could the authors label the anticyclonic and cyclonic eddies that are discussed?*
**Response:** The eddies from Figure 9 are not located in the area of Figure 8; they are situated further north. Therefore, it is not possible to indicate them on Figure 8. The black dots (5, ...) represent the daily positions of the ship starting from the day our experiment began.

*L. 370: "obsevations": observations.*
**Response:** changed

*L. 371. Could the authors correct the sentence?*

**Response:** corrected